# One million years of glaciation and denudation history in west Greenland

Astrid Strunk[1], Mads Faurschou Knudsen[1], David L. Egholm[1], John D. Jansen[2], Laura B. Levy[1], Bo H. Jacobsen[1] & Nicolaj K. Larsen[1,3]

The influence of major Quaternary climatic changes on growth and decay of the Greenland Ice Sheet, and associated erosional impact on the landscapes, is virtually unknown beyond the last deglaciation. Here we quantify exposure and denudation histories in west Greenland by applying a novel Markov-Chain Monte Carlo modelling approach to all available paired cosmogenic $^{10}$Be-$^{26}$Al bedrock data from Greenland. We find that long-term denudation rates in west Greenland range from $>50\,\mathrm{m\,Myr}^{-1}$ in low-lying areas to $\sim 2\,\mathrm{m\,Myr}^{-1}$ at high elevations, hereby quantifying systematic variations in denudation rate among different glacial landforms caused by variations in ice thickness across the landscape. We furthermore show that the present day ice-free areas only were ice covered ca. 45% of the past 1 million years, and even less at high-elevation sites, implying that the Greenland Ice Sheet for much of the time was of similar size or even smaller than today.

[1] Department of Geoscience, Aarhus University, Høegh-Guldbergs Gade 2, Aarhus 8000, Denmark. [2] Institute of Earth and Environmental Science, University of Potsdam, Karl-Liebknecht-Str 24, Potsdam 14476, Germany. [3] Centre for GeoGenetics, Natural History Museum of Denmark, University of Copenhagen, Øster Voldgade 5-7, Copenhagen K 1350, Denmark. Correspondence and requests for materials should be addressed to A.S. (email: astrid@geo.au.dk).

The Greenland Ice Sheet (GrIS) has been waxing and waning through multiple glacial-interglacial cycles over the past few millions of years, thereby sculpting the landscape we observe today. Little is known of the temporal and spatial extent of the GrIS before the last deglaciation, because subsequent erosion has removed most of the evidence. Indeed, the Greenland ice-core records span only the last ~100 kyr (ref. 1), hence the glacial history predating the last glacial cycle remains elusive. As the GrIS advanced onto the continental shelf during the Last Glacial Maximum (LGM), it efficiently eroded deposits from earlier interglacial and glacial periods, leaving only scattered onshore[2–4] and offshore[5–8] fragments of the longer-term geological record. Most Quaternary deposits preserved between the outer coast and the present ice margin have a Lateglacial or Holocene origin, pointing to efficient erosion and a rapid ice retreat across this area during the last deglaciation[9,10]. Over previous glacial cycles, the ice sheet presumably expanded to cover the continental shelf and then retreated inboard of the present ice margin, but well-dated geological evidence testing this hypothesis is lacking.

The origin and age of the ice-sculpted landscapes in Greenland may be studied via quantification of past denudation rates (that is, the removal of mass via physical and chemical weathering), in particular spatial variations in denudation rate. Bedrock erosion rates averaged over the Quaternary in Greenland are relatively unconstrained, but estimated at ~40 m Myr$^{-1}$ based on North Atlantic marine sediment volumes[11,12]. This bulk estimate combines sediment yield from diverse erosion regimes, including diffusive areal scouring and incision focused along valley troughs and fjords. Large parts of the fjord landscapes cut into the west Greenland coastal margin were shaped by selective linear glacial erosion, characterized by large spatial differences in erosion due to variations in the subglacial thermal regime[13]. Here, deep troughs and/or fjords dissect low-relief plateaus that presumably were preserved over multiple glacial cycles under thin and weakly erosive, cold-based ice. The troughs, in contrast, were carved by thick ice masses with basal ice at the pressure-melting point, which facilitates sliding and efficient bedrock erosion[14]. Identifying how and when the present topography of Greenland was established entails quantifying spatial patterns of past denudation rates, which presents a considerable challenge in such landscape settings imprinted by differential thermal glacial regimes[15,16].

Cosmogenic nuclides provide a widely used tool for quantifying landscape denudation rates and exposure history[17]. The method utilizes the constant bombardment of Earth by cosmic rays produced in supernova explosions, which gives rise to secondary cosmic-ray cascades in the atmosphere. When the secondary neutrons and muons penetrate an exposed rock surface, nuclear reactions produce *in situ* terrestrial cosmogenic nuclides (TCNs), of which common nuclides used for dating purposes are [10]Be and [26]Al (ref. 17). Depending on the geological setting, TCNs are commonly used to date the timing of exposure, which in glacial landscapes typically means the last deglaciation, or to constrain the denudation rate of a bedrock surface. Repeated intervals of burial and exposure during successive glacial and interglacial periods cause discontinuous TCN production. This imposes additional difficulties for constraining denudation rates, because the exposure/burial history at a given site is generally also unknown.

In order to accommodate complex exposure histories, paired [10]Be-[26]Al measurements can be used to estimate a minimum-limiting total history and the relative proportion of exposure versus burial, by utilizing that the two nuclides have different half-lives. Under constant exposure, the two nuclides will accumulate following a predictable ratio controlled by the production rates and half-lives, whereas the ratio during periods of burial is governed by the half-lives only[18,19]. This approach, however, has two major shortcomings: It ignores the ongoing erosion and resultant loss of nuclides back through time, and secondly it fails to resolve the alternating nature of exposure and burial through multiple glacial and interglacial periods. Paired [10]Be-[26]Al bedrock data from high-elevation sites in west Greenland and the Baffin Bay area indicate long and complex exposure histories with significant periods of burial, suggesting preservation under weakly erosive, cold-based ice over several glacial cycles[18,19]. In contrast, recently published results from high-elevation surfaces elsewhere in west Greenland (Uummannaq), suggest near-continuous exposure throughout much of the middle and late Quaternary—possibly as nunataks during the LGM and prior glacial maxima[20].

Recent advances in Monte Carlo modelling techniques make it possible to constrain the history of long-term erosion and exposure-burial periods by exploiting more efficiently the differences in production and radioactive decay rates of paired TCNs[21,22]. The Markov-Chain Monte Carlo (MCMC) model approach developed by Knudsen *et al.* (ref. 21) is based on the assumption that the exposure/burial history can be divided into two distinct regimes: (i) glacial intervals with subglacial erosion and, due to shielding by the overlying ice sheet, no exposure, and (ii) interglacial intervals experiencing active subaerial erosion and full exposure, assuming no significant shielding by for example, snow, till, or vegetation (see 'Methods' section). The rates of glacial and interglacial erosion may differ and vary spatially, but for any particular bedrock sample the two erosion rates are uniform throughout all glacials and interglacials, respectively. The MCMC model does not include sudden individual erosion events, such as subglacial plucking, but integrates the effects of plucking events over time. By integrating the glacial and interglacial erosion rates, it is possible to compute a robust, long-term denudation rate for each sample. The exposure/burial history is determined by applying a threshold value to a stacked benthic marine $\delta^{18}O$ record[23], which is a proxy for past global land-ice volume.

In this study, we apply the new MCMC inversion model[21] to all available [10]Be-[26]Al bedrock data from west Greenland, encompassing 49 samples altogether. The most realistic and up-to-date landscape information is integrated as boundary conditions in the model set-up, which enable us to quantify past denudation rates combined with exposure/burial histories. We show that the denudation rate decreases with increasing elevation, from >50 m Myr$^{-1}$ in low-lying areas to 1–5 m Myr$^{-1}$ at high-elevation summit flats (>850 m a.s.l.). We also find that the majority of samples are consistent with presence of an ice cover ca. 45% of the past 1 Myr, whereas the fraction of ice-covered periods was smaller (10–20%) at many high-elevation sites.

## Results

**Denudation rates and landscape evolution in west Greenland.** The paired [10]Be-[26]Al bedrock data derive from four study sites in west Greenland. We apply the MCMC approach to all samples displaying a simple exposure [10]Be age of 20 kyr or more. The simple exposure ages are calculated based on the assumption of continuous exposure, no inherited TCNs, and no post-glacial erosion. The samples with an exposure [10]Be age of 20 kyr or more are believed to violate these preconditions, based on comparisons with ages from boulders and radiocarbon ages of proglacial lake sediments. The relevant samples include 11 samples from Upernavik[19], 19 samples from Uummannaq[20,24,25], 10 samples from Itilleq[26] and 9 samples from Sukkertoppen[20], the two latter sites both belonging to the Sisimiut area (Fig. 1). The

Uummannaq and Upernavik areas have high relief with deep fjords that stretch from the present ice margin to the coast. The samples located close to Sisimiut are from (i) a low-relief area between the fjords (Itilleq), and (ii) from summit plateaus around a local ice cap (Sukkertoppen).

All four sites were sampled across a wide range of elevations from valleys to summits and show an overall trend of increasing simple exposure $^{10}$Be age with altitude (Fig. 2a). From this we identify an elevation threshold, above which all samples display non-negligible inheritance, which must derive from periods of exposure associated with earlier interglacials or ice-free periods. Inheritance is defined by a nuclide inventory that exceeds post-glacial production and implies inefficient bedrock erosion (<2–3 m) over the last glacial period. In reality, however, it is not possible to define a specific threshold below which there with certainty is no inheritance. Our results show that high-elevation summit flats (>850 m a.s.l.) yield the slowest long-term denudation, typically ~1–5 m Myr$^{-1}$, whereas sites at lower elevations have denudation rates of 15–20 m Myr$^{-1}$ (Fig. 2b). There are exceptions to the general trend, as some low-elevation samples have denudation rates <10 m Myr$^{-1}$, demonstrating some spatial variations in the erosion processes shaping the landforms. The overall pattern of decreasing denudation rates with increasing elevation is nonetheless consistent with the distribution of minimum-limiting exposure and burial ages in such landscapes[19,27–29]. This trend reflects that glacial erosion is more efficient in fjords and valleys where the ice is thick enough

to be warm-based and reach the pressure-melting point at the ice-bedrock interface. The low erosion rates found at inter-fjord uplands are consistent with the presence of cold-based ice, which is frozen to the ground and only moves due to internal deformation, thereby preserving the high-elevation areas[30]. The differential erosion has a major influence on the overall evolution of the landscape and is key to understanding the age of landforms and their development through the latter half of the Quaternary. Our analyses further reveal that samples with non-detectable TCN inheritance must have total denudation rates >50 m Myr$^{-1}$ (See Supplementary Data 1). The denudation rates in the fjord troughs (where TCN inheritance is negligible) are therefore in excess of 50 m Myr$^{-1}$ and potentially one or two orders of magnitude higher, as reported by previous studies based on sediment yields[31].

Overall, the denudation rates modelled over the past 1 Myr support the notion of selective linear erosion along the GrIS margins as denudation rates drop more than an order of magnitude with increasing elevation. We cannot exclude the possibility that the spatial patterns of differential erosion are somewhat influenced by bedrock erodibility, caused by for example, variations in bedrock fracture density or orientation[32,33]. The linear appearance and the fairly uniform orientation of the fjords certainly support the notion of a structural inheritance within these glacially eroded landscapes[15].

The low denudation rates at high elevations imply as little as 3–15 m of summit lowering, if extrapolated over the entire

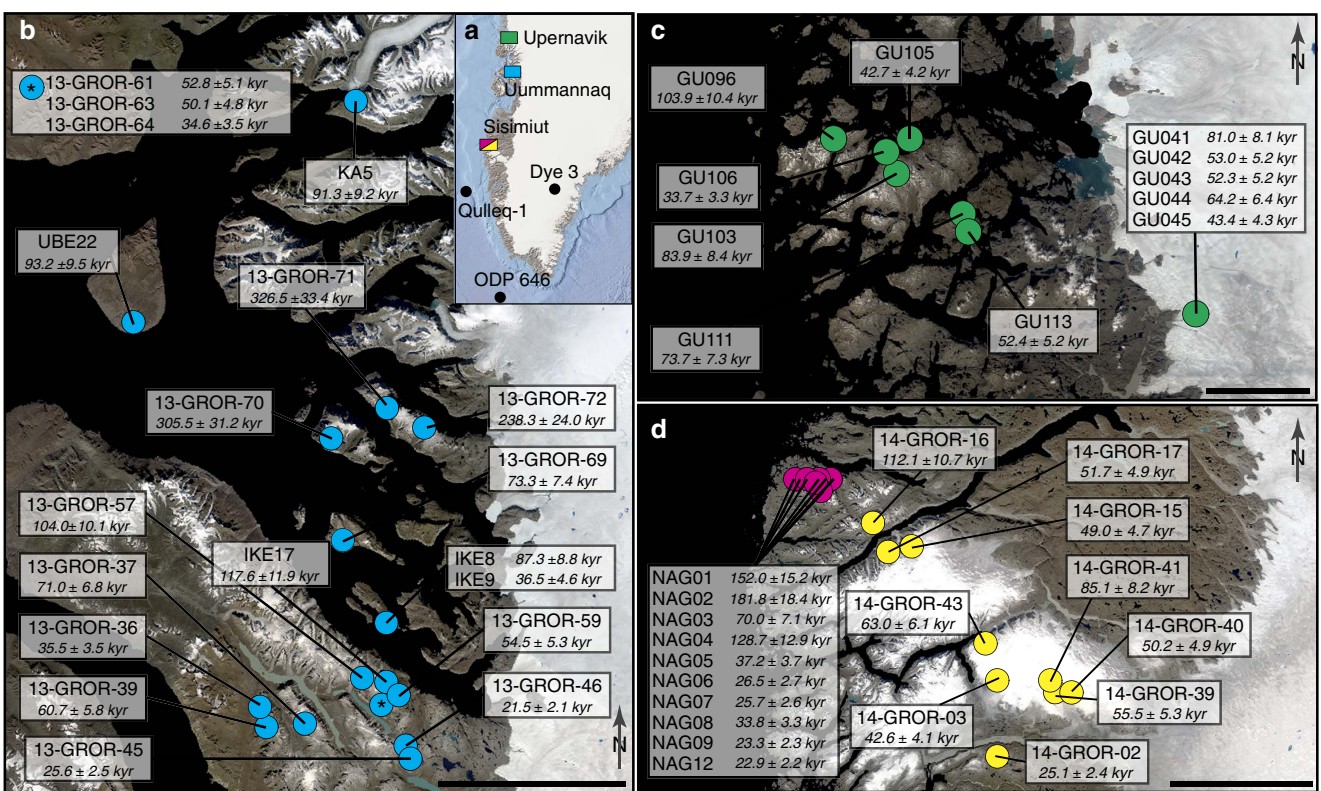

**Figure 1 | Overview maps of sample sites and simple exposure $^{10}$Be ages.** (**a**) South Greenland overview with coloured boxes marking the four TCN sample sites and black circles marking points of pre-LGM Quaternary data[5,7,36]. (**b**) Uummannaq area with sample sites containing $^{10}$Be and $^{26}$Al bedrock data[19]. The point marked with an asterisk represents 3 sample sites, which are shown in the top left box. The scale bar is 50 km wide. (**c**) Sample sites with $^{10}$Be and $^{26}$Al bedrock data[20,24,25] from the Upernavik area. The scale bar is 25 km wide. (**d**) Sisimiut area, covering samples from two sites; the inter-fjord site Itilleq[26] (red) and samples from the margins of the local ice cap Sukkertoppen[20] (yellow). The scale bar is 50 km wide. The data points shown in (**b–d**) meet the criteria of being applicable to the MCMC model approach, by having simple exposure $^{10}$Be ages above 20 kyr and a $^{26}$Al/$^{10}$Be ratio below 7.5. (**b–d**) Simple exposure $^{10}$Be ages are calculated using Cronus Version 2.3 (ref. 45) and the Lal (1991)/Stone (2000) scaling scheme[40,41]. Figure 1 was created using QGIS software[46]. All satellite images are from Landsat8, August 2016, courtesy of the U.S. Geological Survey.

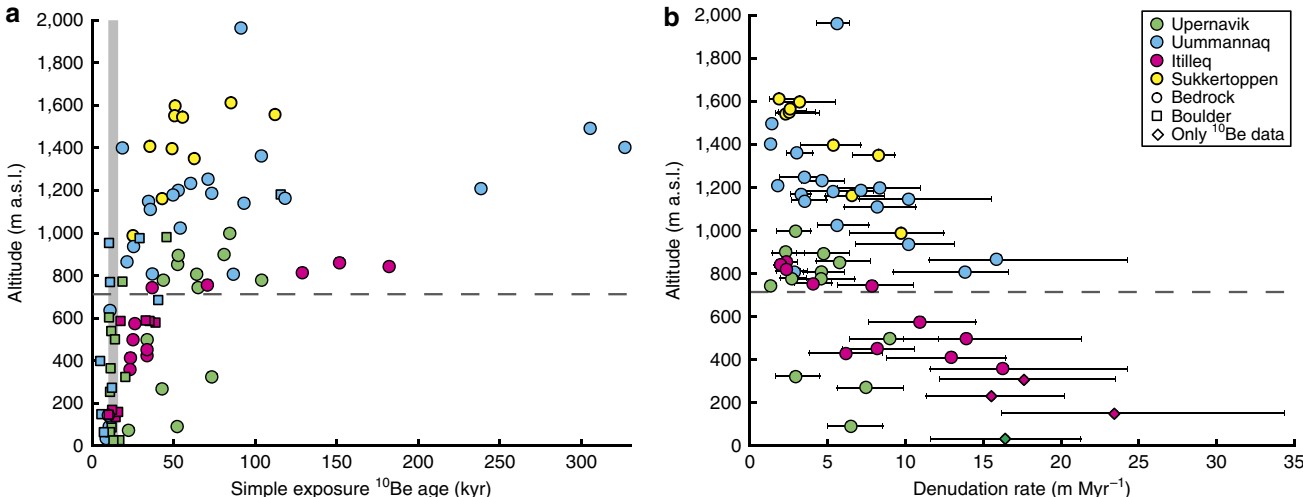

**Figure 2 | Simple exposure [10]Be ages and long-term denudation rates as a function of altitude.** (**a**) Simple exposure [10]Be ages tend to increase with elevation. Above a certain threshold (dashed line), all bedrock samples contain a cosmogenic signal inherited from periods before the most recent glaciation. The grey zone marks the limit used to constrain the timing of the Holocene deglaciation (10–16 kyr) for all sites (see 'Methods' section). [10]Be ages are calculated using Cronus Version 2.3 (ref. 45) and the Lal (1991)/Stone (2000) scaling scheme[40,41]. (**b**) Total denudation rates over the last 1 Myr based on application of the MCMC approach to samples with a simple exposure [10]Be age above 20 kyr and a [26]Al/[10]Be ratio below 7.5. All three sites in west Greenland show a clear trend of decreasing denudation rate with increasing elevation. Diamonds represent samples from the same previous studies, but with information from [10]Be only (that is, no [26]Al data were available). The rates of denudation associated with these samples have larger uncertainties. Error bars are defined as the first and third quartiles of the 200,000 iterations per sample.

Quaternary period. However, we consider this to be a minimum estimate, as the patterns and pace of glacial erosion may have changed systematically over time. Specifically, we envisage that the erosional contrast between high surfaces and glacial troughs has grown over time in response to the topographic change, as the increasing relief around emerging fjords steered more and more ice into the fjords and away from the inter-fjord uplands, which furthermore experienced isostatic uplift[14]. Erosion may therefore have been more uniformly distributed (that is, faster at high surfaces and slower in troughs) during the early glaciations and before the deep fjords were formed. Our MCMC model cannot capture such transitions, as it assumes that rates of glacial and interglacial denudation are uniform in time. The MCMC model does, however, illuminate a robust and systematic elevation-dependence of denudation rates within the most recent glaciations.

**Glaciation history in West Greenland**. The quantification of denudation rates is tied to an estimate of the most likely exposure history, defined by the $\delta^{18}$O threshold for each sample. The exposure histories make it possible to calculate the cumulative sum of exposure time over the alternating ice-free and ice-covered periods over the past 1 Myr. This cumulative, simulated exposure/burial history is conceptually more advanced than the minimum-limiting exposure and burial durations defined by the simplest pathway to explain a point on the two-isotope diagram (Fig. 3a), because it takes into account the most likely timing of ice-free and ice-covered periods based on a proxy for past global ice volume. For all samples, it is possible to define an exposure-burial history that is consistent with the measured [10]Be and [26]Al data. The proportion of cumulative exposure during the last 1 Myr, defined by the $\delta^{18}$O threshold value, varies from <15% to >90%. There is no obvious relationship between the proportion of cumulative exposure time and elevation, but samples characterized by a relatively low degree of cumulative exposure (<25%), or by a high degree of cumulative exposure (>75%),

tend to derive from sites at high elevations relative to the surrounding topography (Fig. 3b). As expected, the cumulative exposure time is closely linked to the [26]Al/[10]Be ratio, with high [26]Al/[10]Be ratios corresponding to high proportions of cumulative exposure (Fig. 3c). In general, the uncertainties associated with estimates of cumulative exposure proportions are relatively high, but they are considerably smaller for samples with a high degree and, in particular, low degree of exposure. For samples with a low degree of cumulative exposure, it is only possible to simulate [26]Al/[10]Be ratios as low as the measured ratios if the exposure is limited to short intervals during the warmest interglacial periods. It is possible, however, that the low-ratio samples were exposed for longer periods of time before the last 3–4 glacial cycles, but such a scenario is beyond the present modelling capability of the MCMC approach, as it requires non-uniform erosion rates and/or a time-dependent $\delta^{18}$O-threshold level.

**Discussion**
The notion of high-elevation surfaces around Uummannaq that remained free of ice during the LGM and earlier glacial maxima was born from high [26]Al/[10]Be ratios indistinguishable from the production ratio of ~6.75 (ref. 20). Here, we demonstrate that the paired nuclide data from Uummannaq and Sukkertoppen displaying high [26]Al/[10]Be ratios are fully consistent with burial during glacial maxima, including an LGM ice cover lasting 15–20 kyr (Fig. 3c). Due to the uncertainty on the measured TCN concentration, it is not possible to firmly establish whether these high-elevation surfaces were ice covered or ice free during the LGM and earlier glacial maxima, as ice-free conditions also remain a possibility for samples with [26]Al/[10]Be ratios indistinguishable from the production ratio. However, the majority (39 out of 49) of the samples point to all three areas being ice covered during glacial maxima. On the basis of the cumulative exposure histories of the samples from Sisimiut, Uummannaq, and Upernavik, we constrain the most likely exposure-burial history associated with the waxing-waning GrIS in these three

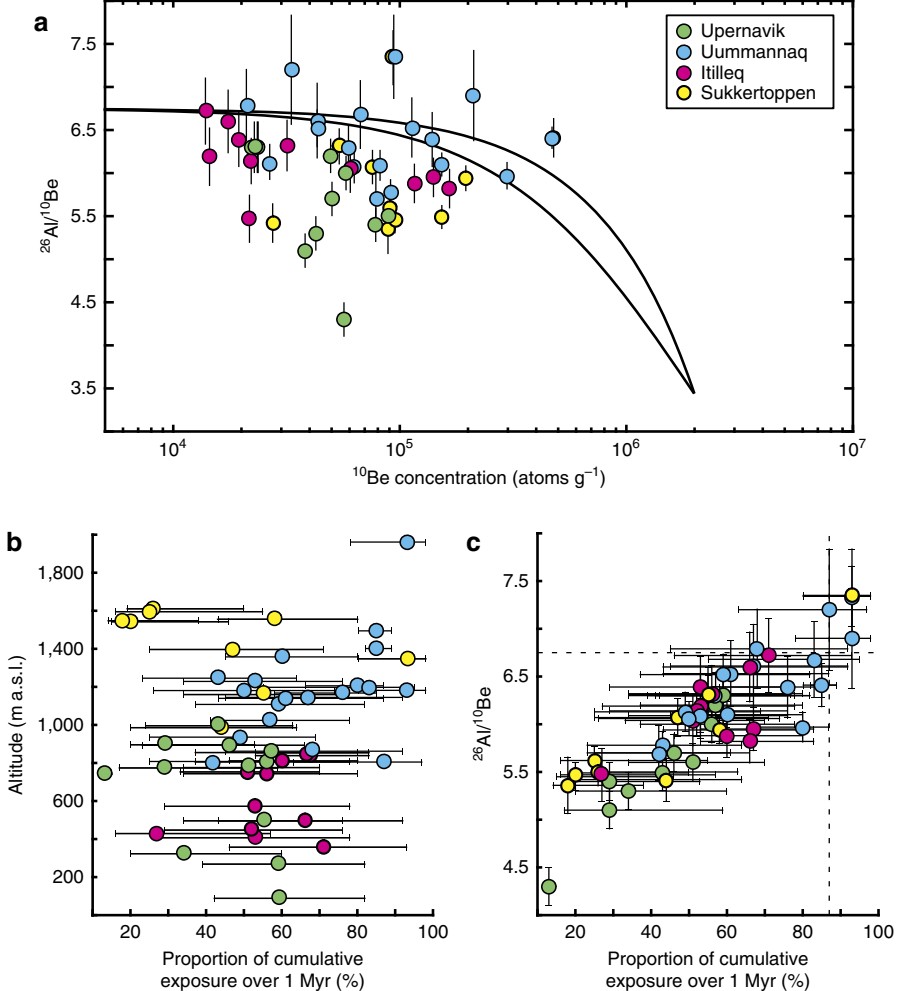

**Figure 3 | Linking nuclide concentrations to exposure history.** (**a**) Plot of $^{26}$Al/$^{10}$Be ratio against $^{10}$Be concentration for all samples used in the MCMC-based quantification of complex exposure histories. The upper black line marks the development of the ratio under constant exposure and the lower black line marks the end points of an infinite number of different steady-state erosion rate scenarios with constant exposure. Error bars are based on uncertainties of the $^{26}$Al and $^{10}$Be concentrations. (**b**) Sample altitude plotted against percentage of time exposed over the last 1 Myr for each sample. (**c**) Plot of $^{26}$Al/$^{10}$Be ratio against percentage of time exposed over the last 1 Myr for each sample. There is a clear correlation between $^{26}$Al/$^{10}$Be ratios and the percentage of exposure over the last 1 Myr based on the MCMC-modelled exposure history. The horizontal dashed line marks the production ratio of $^{26}$Al/$^{10}$Be and the vertical dashed line marks 87% of exposure in 1 Myr, which corresponds to 15 kyr of ice cover during the LGM. Error bars are defined as the first and third quartiles of the 200,000 iterations per sample.

areas. We first compute the most likely exposure-burial history for samples with a simple exposure $^{10}$Be age above 20 kyr and $^{26}$Al/$^{10}$Be ratios lower than the production ratio (See Supplementary Data 1), and we apply Chauvenet's criterion[34] to exclude outliers (for example, samples GU110 and NAG11). This provides an estimate of the broad-scale behaviour of the GrIS at Sisimiut, Uummannaq and Upernavik (Fig. 4), respectively, which on average suggests ice-free conditions for ~55% of the last 1 Myr. On the basis of our analysis of exposure/burial histories, we propose that the expanding GrIS did not engulf these areas in west Greenland when Marine Isotope Stage (MIS) 5e terminated and was succeeded by the subsequent Weichselian/Wisconsin glacial period. The GrIS appears to have expanded across these three areas at around the onset of MIS 4 (~72 kyr ago). The average exposure/burial histories, based on samples with $^{26}$Al/$^{10}$Be ratios indistinguishable from the production ratio (Fig. 4), suggest that most of the high-elevation surfaces around Uummannaq were ice free for almost 90 kyr before the LGM, during which the high surfaces on average may have been ice covered for ~18 kyr. Considerable

spatial variation is indicated by a few sites only experiencing ice-free conditions during peak interglacial periods (for example, GU110). However, in light of uncertainties, we advise some care: the notion of ice advance and retreat occurring everywhere in the landscape simultaneously is overly simplistic. At some sites, the exposure-burial histories were possibly further complicated by partial shielding associated with thin ice covers, meaning that the TCN production was not completely halted during glacial periods. The presence of such thin overlying ice covers would prolong the estimated duration of ice-covered periods, but exert negligible effect on the estimated erosion rates.

Our results reveal a glaciation history for west Greenland that is in accordance with the sparse geological evidence available, which suggests ice-free conditions in the fjord area for the Holocene, the early Weichselian/Wisconsin, and during MIS 5e (ref. 2). We find that ice-free periods associated with interglacials MIS 5e, MIS 11 and MIS 21 stand out in all three areas as longer than other ice-free intervals. Based on a reconstruction of Arctic temperatures from a Siberian lake, MIS 11 is considered significantly warmer than other interglacials[35]. Therefore, if the

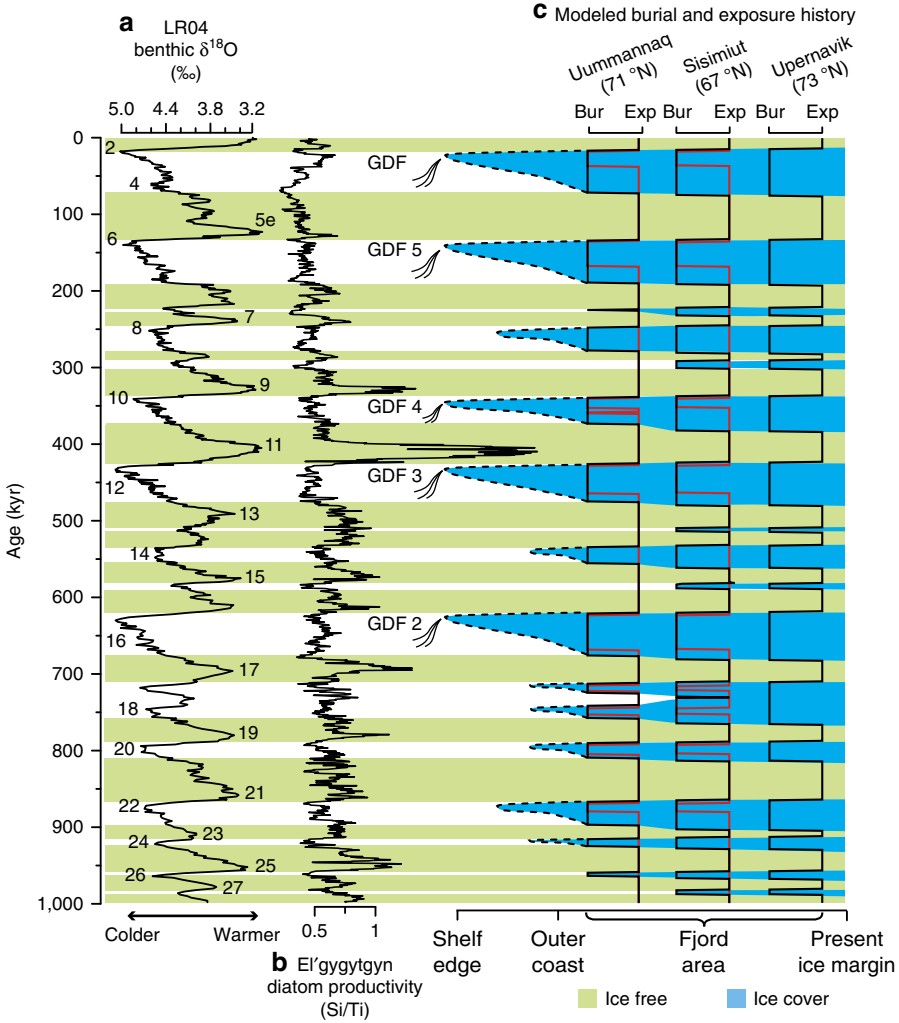

**Figure 4 | Climate and modeled glaciation history in west Greenland throughout the last 1 Myr.** (**a**) The stacked benthic marine δ[18]O record[23] is a proxy for global ice volume and is divided into numbered marine isotope stages (MIS). We determine the exposure history by applying a threshold to this global climate record. (**b**) The diatom productivity curve from Lake El'gygytgyn[35], which is a temperature proxy based on limnic Arctic data, indicating very warm conditions during MIS 11. (**c**) Quantification of the most likely periods of exposure ('Exp' and green coloration) and burial by ice cover ('Bur' and blue coloration) based on 39 bedrock samples (black lines) from four sites in the Fjord Area of west Greenland[19,20,24–26] (the Sisimiut exposure/burial history covers the sample sites Itilleq and Sukkertoppen). Red lines mark the glaciation history of samples that only have experienced very short durations of burial and therefore have [26]Al/[10]Be ratios at or above the production ratio of 6.75. The red lines are based on three samples from Sisimiut and seven samples from Uummannaq. It is likely that the ice sheet extended to the shelf edge and deposited glacial debris flows during the five most prolonged ice-covered periods[5,37].

interior of southern Greenland was completely deglaciated, as geological findings suggest[7,36], it most likely happened during MIS 11, which appears to have been both warm and relatively long-lasting. Our results also indicate that the ice-covered periods with the longest duration coincide with deposits of marine glacial debris flows in the Davis Strait[5,37] (Fig. 4). Four of these five long-lasting ice-covered periods, where the ice extent reached the Greenland shelf, most likely experienced > 50 kyr of continuous ice cover, except for most high-elevation plateau locations that probably were ice-covered only during glacial maxima (Fig. 4c, red line).

In summary, we illustrate how multiple paired cosmogenic nuclides can be used to shed light on the response of the GrIS to climate change during the last 1 Myr. We constrain the most likely glaciation histories and differential denudation rates in the fjord areas of west Greenland, and hereby quantify rates of lowering associated with various glacial landforms. Our application of the MCMC modelling approach to paired cosmogenic nuclides opens new avenues for quantifying glacial and

interglacial denudation rates, which are essential to understand long-term landscape evolution and the origin of the glacial landscape we observe today.

## Methods

**Markov-Chain Monte Carlo inversion approach and model set-up.** The novel MCMC inversion model[21] used in this study to quantify past denudation rates and exposure/burial histories includes four model parameters; the interglacial and glacial erosion rates (in m Myr[−1]), as well as the timing of the last deglaciation (in kyr) and the δ[18]O-threshold level (in ‰). Application of the δ[18]O-threshold level to the global benthic marine δ[18]O record[23], which is a proxy for changes in past global ice volume, is used to define the exposure/burial history in the model simulations, following the idea presented by Stroeven et al.[38]. The stacked δ[18]O record is smoothed using a 5-kyr running window so it reflects the major marine isotope stages (MISs) and sub-stages. The smoothed δ[18]O record thus allows changes in exposure/burial regimes in the model set-up that are consistent with the available knowledge of large-scale glacial advances and retreats in Scandinavia[30]. The Holocene deglaciation is a free parameter in the model and we apply a set of wide boundary values, allowing the deglaciation to take place between 10–16 kyr in all three areas. This is a rather conservative estimate based on bedrock-boulder pairs from each of the three areas[10,19,24–26] and allows considerable variation due to elevation and distance to present ice margin. The interval of the Holocene

deglaciation also frames the possible range of $\delta^{18}$O-threshold values, which are allowed to vary between 3.6‰ and 4.7‰. For samples with $^{26}$Al/$^{10}$Be ratios indistinguishable from the production ratio of ca. 6.75, we expand the $\delta^{18}$O boundary values to 3.6–5.0‰, allowing for the possibility of continuous exposure throughout the last 1 Myr. The glacial and interglacial erosion rates have broad boundary values of 0.1–1,000 m Myr$^{-1}$. We used all available paired $^{26}$Al/$^{10}$Be data for bedrock samples in Greenland with a simple exposure $^{10}$Be age above 20 kyr and a $^{26}$Al/$^{10}$Be ratio below 7.5. Ratios above this level are unlikely to form in any burial and/or exposure scenario and they cannot be simulated based on the currently known muonic and spallogenic production rates.

We follow the procedure demonstrated by Knudsen et al.[21] and compute the production and decay/erosion of $^{10}$Be and $^{26}$Al throughout the Quaternary for different combinations of the model parameters, and compare the results to the measured concentrations between each simulation. We use a sea level high-latitude $^{10}$Be production rate of 4.01 atoms g$^{-1}$ per year and $^{26}$Al production rate of 27.07 atoms g$^{-1}$ per year, based on the calibration set by Borchers et al.[39] for $^{10}$Be and $^{26}$Al surface production rates and the Lal (1991)/Stone (2000) scaling scheme[40,41] for all 49 samples from the three areas in west Greenland (see Supplementary Data 1).

The Metropolis-Hastings MCMC technique[42,43] is used to map the family of model parameters that provides the best, weighted least-squares fit to the measured data. For each sample, we use four 'random walks', which start at different places in the model space, to ensure that the result does not depend on the starting position of the search through the model space. A burn-in phase of 1,000 iterations is used to make a crude initial search of the model space, whereas 50,000 iterations and an acceptance ratio of 0.4 are used in the main MCMC phase, when finding the most probable scenarios amongst the $4 \times 50{,}000$ iterations. To estimate the model parameters for each walker, we use the median of the 200,000 simulations, whereas the associated uncertainties are based on the 25% and 75% quartiles.

**Compilation of the model output.** As the results obtained with the four different walkers are very similar, we use the average of all simulations for each sample (200,000) to estimate the model parameters, which, in turn, makes it possible to estimate the total denudation rate as well as the exposure/burial history. The exposure/burial history associated with each sample emerges by applying the median $\delta^{18}$O threshold value (Supplementary Fig. 1) to the global marine benthic $\delta^{18}$O record[23], hereby defining periods of exposure and burial as the intervals below/above this threshold, respectively. Supplementary Fig. 2 shows an example of the $\delta^{18}$O threshold value for one sample. The exhumation history and total denudation rate over the last 1 Myr are estimated for each sample by integrating the glacial and interglacial erosion rates (Supplementary Fig. 1) with the burial/exposure history, and subsequently computing the median exhumation history based on the 200,000 simulations (example of exhumation history is shown in Supplementary Fig. 3). The minimum denudation rate of $>50$ m Myr$^{-1}$ computed for low-lying areas and glacial troughs are based on 25% quartiles from samples with uncertainties that overlap the timing of the Holocene deglaciation. We compile the exposure/burial history for each of the three areas (Supplementary Fig. 4) by taking the average median $\delta^{18}$O threshold value of all samples with a simple exposure $^{10}$Be age exceeding 20 kyr. The samples within each area are additionally grouped into samples with $^{26}$Al/$^{10}$Be ratios below the production ratio (6.75) and samples with ratios above, or indistinguishable from, 6.75.

**Till-cover sensitivity.** We test the effect of a till cover during exposure periods, which would dampen the nuclide production and could cause too-young exposure ages if not taken into account. To understand how the reduced nuclide production due to the presence of a till cover with variable thickness affects the estimate of the denudation rate and $\delta^{18}$O-threshold value, we apply the correction factor for till using equation (1) (ref. 17):

$$f_{\text{till}} = e^{-z\rho\Lambda^{-1}} \tag{1}$$

to the nuclide production rates, using a till density of 2,200 kg m$^{-3}$ (ref. 44). Supplementary Fig. 5a–f shows how four different samples (GU041, GU113, 14-GROR-40 and 13-GROR-70) respond to till with a thickness varying from 0.1 m to 1.0 m covering the bedrock during 25, 50 or 100% of the ice-free periods. The samples derive from three different sites and represent different elevations and landscape settings. The lowermost points in each subplot show the modelled denudation rate (Supplementary Fig. 5a–c) and $\delta^{18}$O-threshold value (Supplementary Fig. 5d–f) without till cover.

In general, the $\delta^{18}$O-threshold value is less affected by till cover than the denudation rate, but the effect is indistinguishable with regard to both denudation rate and exposure history (determined by $\delta^{18}$O-threshold value) for most of the samples. Except for one sample, it is only in the most extreme cases of 0.5–1.0 m of till cover during 100% of the ice-free periods that the results are significantly affected by the presence of till. The presence of a till cover would result in denudation rate estimates that are slightly too high and $\delta^{18}$O-threshold value estimates slightly too low, if the till cover is not accounted for. The magnitude of these effects depends on the duration and thickness of the till cover. In general, till covers are rarely observed today in the sampled regions, and we do therefore not expect till cover to present a significant problem when estimating the landscape history in west Greenland based on TCNs.

**Data availability.** The authors declare that the main data supporting the findings of this study are available within the paper and its Supplementary Information files. Extra data are available from the corresponding author on request.

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

## Acknowledgements

M.F.K. and N.K.L thank the Villum Foundation's Young Investigator Programme for their support. J.D.J. was supported by a Marie Skłodowska-Curie Fellowship. D.L.E. was supported by the Danish Council for Independent Research. We thank Lee Corbett and Arjen Stroeven for highly constructive and helpful reviews.

## Author contributions

A.S., N.K.L. and M.F.K. designed the study and A.S. performed the calculations. D.L.E., B.H.J. and L.B.L. helped interpreting the results. A.S., N.K.L. and M.F.K. wrote the paper with contributions from D.L.E. and J.D.J.

## Additional information

**Competing financial interests:** The authors declare no competing financial interests.

