## [Peer Review File · Nature Communications]

Reviewers' comments:

Reviewer #1 (Remarks to the Author):

Review of manuscript by Strunk and others for consideration in Nature Communications

In the manuscript provided by Astrid Strunk and her colleagues they use the information contained in cosmogenic nuclides of bedrock samples in western Greenland to estimate total site burial and exposure histories given a range of interglacial and glacial erosion rates, deglaciation durations and oxygen isotope cut-off values applied to a 1 Myr record of glaciation from deep-sea records. The results are quite spectacular in their precision but they also confirm spatial patterns of erosion that have been shown using cosmogenic isotopes from other formerly glaciated margins (such as in North America and in Scandinavia) and that were postulated for the Greenland margin from geomorphology (Sugden, 1974; Sugden, D. E. (1974). "Landscapes of glacial erosion in Greenland and their relationship to ice, topographic and bedrock conditions." Institute of British Geographers Special Publication 7: 177-195). Indeed, in a paper by the same group in Quaternary Geochronology (Knudsen et al., 2015), they already show the range of exposure and burial durations and erosion/denudation rates for the western Greenland margin as presented in the current manuscript for a complete dataset for this region. Hence, whereas the paper is cutting-edge and novel, it represents more of a modest step in a new direction than is acknowledged in the current manuscript. For example, it would be good to discuss why current results for samples GU110 and GU111 differ from those of Knudsen et al. (2015), how their results scale to the total topography which has probably formed over a 7-10 Myr duration (and which informed Sugden to propose a very similar landscape evolution model for this margin) and how their results compare to information from margins, such as onshore geomorphology and cosmogenic evidence from the Scandinavian margin (Kleman and Stroeven, 1997; Fabel et al., 2002; Stroeven et al., 2002 and several subsequent papers by Norwegian and Swedish colleagues).

The applied Markov-Chain Monte Carlo modeling approach has a number of assumptions all of which are clearly presented. However, it remains unclear how sensitive derived conclusions are for uncertainties in the assumptions. The model assumes, for example, subglacial erosion and no exposure for glacial conditions and full exposure and subaerial erosion during ice free periods. Particularly the latter assumption is tenuous as glacial bedrock landscapes are frequently mantled by glacial till. If such surfaces were covered by till for lengthy periods of time, this would probably lead to a current underestimation of interglacial erosion rates (when they occur) than currently expressed? Another assumption, and a difference from previously applied approaches, is the decoupling of the deglaciation age and the oxygen isotope cut-off value. This is intuitively wrong as - except for the Holocene (which is decoupled) - the cut-off value determines ice cover/ice free conditions for a site for the rest of the 1 Myr history. An inclusion of sensitivity analyses regarding model parameters appears to be required for the current manuscript.

I am not impressed with the focus on MIS 11. The argument that it was of longer duration than other ice free periods is one which has been raised many times before and this manuscript adds little to that. Rather, I would want to motivate the authors to use the ink to put their interesting results in a broader context.

I much like the manuscript of Strunk and colleagues and would endorse publication given a broader background description (context), more in-depth discussion of their results against previous literature (it is particularly troublesome that Sugden is missing from the list of references), abandon their MIS 11 focus, and an inclusion of sensitivity tests. The manuscript is well written, is based on a methodology which has been presented in detail elsewhere (Knudsen et al., 2015) and which appears robust, and has some spectacular quantitative results. I conclude with a few short comments:

1. Mention of ODP646, Qulleq-1, and ice cores in Fig 1 caption.
2. Line 337, remove "to" between "model" and "the samples".
3. Figure 2: why is the dashed horizontal red line in-between two red dots? Why does the upper one have inheritance and the lower one not?
4. Supplementary line 37: a value "below" (i.e. lower than) the cut-off value means more interglacial not "glacial"!

5. Supplementary line 60 and 69-71: UBE 13 is evidently an oddball. Was it not removed from mean values and thus Uummannaq mean value is perhaps based on 5 samples rather than 6?
6. Reference quoted and not used in either this ms or in Knudsen et al. (2015): Stroeven, A. P., et al. (2002). "Quantifying the erosional impact of the Fennoscandian ice sheet in the Torneträsk-Narvik corridor, northern Sweden, based on cosmogenic radionuclide data." *Geografiska Annaler* 84A(3-4): 275-287.

Reviewer #2 (Remarks to the Author):

Review of:

One million years of glaciation and denudation history in Greenland
by Strunk et al.
Nature Communications, June 2016

In this manuscript, the authors use a numerical model to quantify rates of denudation and the timing of exposure/burial on the west Greenland landscape. They make use of several existing cosmogenic nuclide datasets (those with ^{10}Be and ^{26}Al analysis on bedrock surfaces) as their data source. They employ an advanced Monte Carlo modeling approach that uses certain inputs (interglacial and glacial erosion rates, oxygen isotope threshold that results in glaciation of the study areas) to infer the most likely scenarios for the study surfaces. The authors conclude that fjord areas have more rapid denudation rates than the highlands, and that there were likely several periods of significant interglacial exposure, in particular MIS11.

I will mention that I am by no means a modeling expert, so the Editor may wish to solicit feedback from someone with more quantitative background than myself who can evaluate the methods thoroughly. Here, I primarily focus on the conceptual framework of the study and the interpretation rather than the details of the model.

Overall, I would like to commend the authors on a job well done. These are interesting questions to pursue, and the authors make good use of pre-existing datasets but take those data to new places. The paper is generally well-written and clear, and the significance of the work is effectively communicated in the introduction.

However, I think the paper could benefit from revision, primarily targeted at increasing the level of depth in the discussion, more effectively communicating the complexities associated with the data and resulting interpretations, and making the writing more accessible to a diverse group of readers. At present, I am not entirely convinced that the paper will be accessible to and/or of interest to a broad audience (although the climatic implications are interesting, and potentially far-reaching). Bringing additional ideas into the discussion, particularly those focused on long-term landscape evolution, may help the paper be of interest to a wider range of readers.

Below, I have detailed several major suggestions as well as a number of minor suggestions. Again, I commend the authors on an interesting study and encourage them to continue expanding its depth and accessibility.

Lee Corbett
Ashley.Corbett@uvm.edu

Big-picture comments:

Knudsen (2015) paper. I was surprised not to see more explicit discussion of the Knudsen (2015)

paper here. Although I admittedly lack modeling expertise, the two papers seem like they share some of the same ideas and utilize some of the same data. My impression is that the 2015 paper outlines the technique, while the in-review paper presents findings based on the technique. Perhaps there was additional discussion in the cover letter that I am missing? I don't necessarily see a problem here, I just feel like I am lacking clarity on how the two are different, how the second builds on the first, and how much of the data in the second is new.

Background information. I understand you have very limited space here, but is it fair to assume that all readers know what cosmogenic nuclides are? Possibly in Quaternary Science Reviews, but probably not in a Nature Communications piece. Your paper would be more easily accessible to a broad audience if you could add a sentence or two about what these nuclides are and how they are produced. Similarly, a sentence about the difference between warm-based and cold-based ice would be helpful to many readers, as would a sentence about the difference between an erosion rate and a denudation rate. Finally, several terms used in the Methods section could be more effectively explained (see detail below).

Model assumptions. You have some good discussion of this on Lines 67-79, although personally I would like to see more detail here since communicating the assumptions is so critical for allowing the reader to understand the model (and its limitations). For example, your model does not give you the ability to include periods of interglacial burial of the bedrock surfaces (i.e. by snowfields or till), which is a process often discussed in the Arctic literature. Nor does your model allow you to partially shield a surface, erode a surface episodically, etc. I don't think this is in any way a problem- all models require assumptions- but a more thorough discussion of this could be beneficial.

Erosion rates. Nowhere in the paper do you mention the interglacial and glacial erosion rates you used and how you determined them. You describe this in the supplement, but to me it seems like an important piece of information that should be made explicit for the reader. I suggest including a sentence to this effect in the paragraph where you discuss the model; at the least, provide the range of values you used, a brief statement of how you determined those values, and refer the reader to the supplement for more information.

Samples with "no" inheritance. In a number of places in the text (Lines 89-91, Lines 105-109, and Figure 2), you identify samples that have "no" isotopic inheritance, often in relation to an inferred topographic boundary. To me, this seems overly simplistic. Can you say with certainty that the low-elevation samples are inheritance-free? And how can you confidently distinguish between which samples contain inherited nuclides and which do not? Realistically, all of the samples in your dataset may have at least small amounts of inherited nuclides, but the uncertainties in the single-isotope simple exposure ages and/or the uncertainties on the $^{26}\text{Al}/^{10}\text{Be}$ ratios do not allow you to distinguish between a sample with no inheritance and one with minimal inheritance. I think this is an important point to finesse throughout the paper, especially since it is a limitation of your model and of the technique as a whole.

Denudation rates. One of your main aims is quantifying the difference in denudation rates between fjords and inter-fjord plateaus, and it seems like you have successfully been able to do this. I would be interested to see you take this a step further though, and tie it back to the landscape in west Greenland today. What is the current elevation difference between the fjords and the highlands? When extrapolated over the lifespan of the GrIS, is that at all consistent with the difference in denudation rates you calculated? If not, could there be structural underpinnings to the origin of the fjords? You probably cannot address this question with a high degree of certainty, but I think it would be interesting discussion to include and would make your study less theoretical and more relevant to understanding the modern-day landscape. At present, your claim of understanding the age and origin of the present topography (Lines 51 and 104) seems unrealized.

"Chicken and egg" problem. I guess this is an American expression, but hopefully my analogy

makes sense. One of your main conclusions is that the fjord areas have higher denudation rates than the intervening highlands, and that this is likely due to the subglacial thermal regime. I agree with your assessment. But are the fjords deep because of the rapid denudation rate? Or is the rapid denudation rate only able to occur because the fjords are deep? Is this a positive feedback loop? In essence, what came first, the topography or the denudation rate? This is a subtlety, but I think it is an interesting question and warrants further exploration. Similar to my point above, this sort of critical discussion will help take your paper from the model to the actual landscape.

Minor comments:

Line 14 and throughout: I usually see Arctic capitalized.

Line 19: I would advise against using "all" here; I know there is at least one more bedrock paired-isotope study that I reviewed, and it might be published in advance of your paper.

Line 24: Do you mean "durations" instead of "periods"? I would use "duration" here to clarify that this is a cumulative amount of time rather than a single period of time.

Lines 26-29: I think you are referring to MIS11 in this sentence, but the usage of "this period" and "this time" is vague.

Lines 39-41: As written, the sentence seems to argue that the age of the deposits (lateglacial/Holocene) implies rapid retreat. I think the sentence might need some restructuring. I assume you mean that there are some specific retreat rate estimates?

Lines 52-60: The end of this paragraph does not read as smoothly as the rest of your front material. I wonder whether calling out the Svalbard study specifically is worth the space in such a short paper format. You may be better served by condensing this down to a concise sentence focused on the limitations of previous approaches.

Lines 76-79: I like that you constrain the timing of the most recent deglaciation with independent radiocarbon and/or cosmogenic chronology. For the sake of methodological transparency, it might be helpful to include a table in the supplement of exactly what ages you used and the studies from which they came. This could help future groups, perhaps those with a different model, to use the same assumptions as you did and compare results.

Lines 81-82 and Lines 207-208: How did you decide upon this 20ka threshold for which samples you modeled? In the second occurrence, you say that this age is "significantly higher than the age of Holocene deglaciation". But did you actually apply some sort of statistical test, or is this a semi-arbitrary cutoff? If the latter, omit the word "significant".

Lines 97-100: The reasoning here seems circular. You say that your calculated denudation rates and calculated exposure/burial durations from previous studies are consistent with one another. But aren't these all based on the same isotopic data, so wouldn't we expect them to be consistent?

Lines 107-108: I think you mean "are therefore" instead of "is therefore".

Line 123: See my earlier comment (Line 24) about "durations" versus "periods".

Lines 123-124: It is not clear what the values in parentheses are. Are these first and third quartiles as mentioned in the methods text? I suggest defining this here so that it is easily accessible for the reader.

Line 127: Using the "GDF" acronym here seems unnecessary since you only use the term in this one paragraph. It might be friendlier to the reader to omit it.

Line 137: Define "Weichselian"; most of the audience probably will not be familiar with this term (or perhaps just use the MIS notation instead).

Lines 161-162: Use "1 Myr" or "one million years".

Line 209: I think you mean "for" instead of "four".

Line 209 and below (and throughout the supplement as well): What is a "walker"? I am not sure this is a term most of the audience will be familiar with.

Line 212: What is the "acceptance ratio"? Similar to the above, this probably will not be familiar to non-modelers.

In-text references: It looks like there are numerous instances where the references are not superscripted. See lines 118, 128, 139, 148, 152, and 155.

Figure 1. I think this figure could be improved to convey more information. The base imagery here really is not great (and the source of the imagery should be included in the figure caption). Is there another source of imagery that is higher resolution? Additionally, you could contemplate adding some quantitative information (perhaps simple ^{10}Be exposure ages, or maybe dot size as scaled to exposure age?).

Figure 3. It seems odd to me to graph the exposure history on a scale of 0-100% when there are only ever two possible values: 0 or 100. Is there a more intuitive way of symbolizing this for the reader that makes the fact that this is a bimodal system (either complete exposure or complete burial) more clear? Showing this another way might also help the legibility of the plot, since these lines are hard to read because of the underlying colors and the fact that they are close together. Are the plots even necessary, or can you just say in the caption that the study sites are exposed in the green intervals and buried during the blue intervals?

All figures. The way in which you denote sub-elements of the figure (a, b, c) is not standard throughout. In some, you lead with the letter; in others, the letter follows the description. Standardizing your notation will make these easier to read.

Reviewers' comments:

Reviewer #1 (Remarks to the Author):

Review of manuscript by Strunk and others for consideration in Nature Communications

In the manuscript provided by Astrid Strunk and her colleagues they use the information contained in cosmogenic nuclides of bedrock samples in western Greenland to estimate total site burial and exposure histories given a range of interglacial and glacial erosion rates, deglaciation durations and oxygen isotope cut-off values applied to a 1 Myr record of glaciation from deep-sea records.

The results are quite spectacular in their precision but they also confirm spatial patterns of erosion that have been shown using cosmogenic isotopes from other formerly glaciated margins (such as in North America and in Scandinavia) and that were postulated for the Greenland margin from geomorphology (Sugden, 1974; Sugden, D. E. (1974). "Landscapes of glacial erosion in Greenland and their relationship to ice, topographic and bedrock conditions." Institute of British Geographers Special Publication 7: 177-195).

Response: We agree with this overall assessment. The spatial patterns of erosions have been discussed in several earlier studies based on cosmogenic nuclide data and other lines of evidence. The main contribution of this paper is that we quantify erosion rates in a framework that also takes into account the most likely exposure-burial history.

As we were revising the paper, Beel et al. (2016) published a new study presenting paired $^{26}\text{Al}/^{10}\text{Be}$ bedrock data from west Greenland (23 samples). We have included the new $^{26}\text{Al}/^{10}\text{Be}$ bedrock data in our study, and strived to place our results in a broader context, as suggested below.

Indeed, in a paper by the same group in Quaternary Geochronology (Knudsen et al., 2015), they already show the range of exposure and burial durations and erosion/denudation rates for the western Greenland margin as presented in the current manuscript for a complete dataset for this region. Hence, whereas the paper is cutting-edge and novel, it represents more of a modest step in a new direction than is acknowledged in the current manuscript. For example, it would be good to discuss why current results for samples GU110 and GU111 differ from those of Knudsen et al. (2015), how their results scale to the total topography which has probably formed over a 7-10 Myr duration (and which informed Sugden to propose a very similar landscape evolution model for this margin) and how their results compare to information from margins, such as onshore geomorphology and cosmogenic evidence from the Scandinavian margin (Kleman and Stroeven, 1997; Fabel et al., 2002; Stroeven et al., 2002 and several subsequent papers by Norwegian and Swedish colleagues).

Response: We understand this point concerning the novelty of the results, but there is a huge difference between this paper and the results presented in Knudsen et al. (2015). The paper by Knudsen et al., 2015 presented the Markov-Chain Monte Carlo modeling approach to paired multi-nuclide data. The main point of this paper was to demonstrate the concept/model, the robustness of the approach via examples involving synthetic scenarios, and the relevance by applying the model approach to real data. To demonstrate the relevance, Knudsen et al. (2015) applied the model to two arbitrary samples from Upernavik (from Corbett et al., 2013), but no consideration was given to the boundary conditions of the model. For instance, Knudsen et al., used a 30-kyr smoothing window to

the benthic $d^{18}O$ record of Lisiecki et al., whereas we use a 5-kyr smoothing window. The latter approach applied in this study provides a much more realistic glaciation history, in which MIS 5a and MIS 5c are clearly distinguishable (which they aren't for a 30-kyr smoothing window). The resulting $d^{18}O$ curve, which is used to define the exposure-burial history - is much more in line with evidence from Scandinavia (Mangerud et al.) and the $d^{18}O$ record used in studies of the exposure-burial history in Scandinavia based on ^{26}Al and ^{10}Be (e.g. Fabel et al., EPSL, 2002). We also use more realistic boundaries for the timing of the last deglaciation, and the final results presented in this study, i.e. the denudation rates and exposure-burial histories, are therefore much more realistic. It was never the purpose of Knudsen et al. (2015) to present and discuss detailed results in detail, but rather to demonstrate that the method was applicable to real data.

The present manuscript is profoundly different from Knudsen et al. (2015), as we apply the MCMC model approach to all available paired multi-nuclide bedrock data from Greenland – including the two samples from Upernavik (GU110 and GU111). Here, we quantify denudation rates and exposure histories based on data from 49 samples, and strive to extract as much information on the regional landscape evolution as possible. In this study, we also use the latest production rates and muon attenuation length scales. These factors combined with the wider boundaries associated with the timing of the last deglaciation and – in particular – the different degree of smoothing of the $d^{18}O$ record lead to slightly different results (and larger uncertainties) for samples GU110 and GU111. We have chosen not to discuss these discrepancies in the present paper, but we do briefly highlight the difference between this study and that of Knudsen et al. (2015).

In the revised paper, we have additionally incorporated a discussion of our new results in the context of the long-term topographical evolution. We also provide a broader background description, which include the selective linear erosion model of Sugden (1974).

The applied Markov-Chain Monte Carlo modeling approach has a number of assumptions all of which are clearly presented. However, it remains unclear how sensitive derived conclusions are for uncertainties in the assumptions. The model assumes, for example, subglacial erosion and no exposure for glacial conditions and full exposure and sub-aerial erosion during ice free periods. Particularly the latter assumption is tenuous as glacial bedrock landscapes are frequently mantled by glacial till. If such surfaces were covered by till for lengthy periods of time, this would probably lead to a current underestimation of interglacial erosion rates (when they occur) than currently expressed? Another assumption, and a difference from previously applied approaches, is the decoupling of the deglaciation age and the oxygen isotope cut-off value. This is intuitively wrong as - except for the Holocene (which is decoupled) - the cut-off value determines ice cover/ice free conditions for a site for the rest of the 1 Myr history. An inclusion of sensitivity analyses regarding model parameters appears to be required for the current manuscript.

Response: We agree that the presence of till, vegetation, or snow cover could have an effect on our denudation rate or exposure history results - as it would have on any TCN dating results from such areas. Therefore, we now include a sensitivity analysis of till cover in the interglacial periods, where we apply a till cover of thicknesses from 0.1m to 1.0m in 25% to 100% of the time of each interglacial. The analysis is done for four samples, representing all three regions and from different elevations. We find that only in the most extreme case of 1.0m of till cover during 100% of the interglacials, the till cover has a significant effect on the denudation rate and exposure history, compared to the case with no till cover at all. However, we consider it unlikely that the sampled

sites included in this study were affected by a significant till cover, as the exposed bedrock appears completely free of till or any other permanent cover (including). To our knowledge, there is no evidence suggesting that the glaciers deposited a till cover at high elevations anywhere in Greenland. Moreover, we don't see how a significant till cover would have been removed if it was present earlier in the Holocene on these high-elevation summit flats. If there was significant till cover, we would expect bedrock ages to be younger than boulder ages on average, but that is not the case for the TCN data included in this study.

We understand why the decoupling of the deglaciation age and the oxygen isotope cut-off value may seem intuitively wrong. We only partially agree with this statement, however, because the deglaciation in some cases is very well defined and the assumption that all earlier glacial-interglacial transitions occurred at exactly the same $d^{18}O$ value seem unrealistic – and quite a strong assumption. Decoupling the two parameters allow for some flexibility concerning when the glacial-interglacial transitions occurred in the past compared to the onset of the present ice-free period. Importantly, when the two most important exposure periods contributing to the present TCN inventory (the Holocene and MIS 5e – the earlier exposure periods are tied to MIS 5e, but their contribution to the present TCN inventory is significantly smaller) are decoupled and may thus inform us on the possible exposure histories that are consistent with the data (as revealed by the MCMC model).

However, in light of the new data included in this study (the recent publication by Beel et al., 2016) and the review comments above, we changed the boundary values for the deglaciation and the $d^{18}O$ cut-off threshold. We now let the boundary conditions of the oxygen isotope cut-off value be determined by the boundaries of the deglaciation age, which we conservatively set to 10-16 kyr for all three areas. The wider deglaciation boundary values cause slightly larger error bars, but the results remain unchanged.

I am not impressed with the focus on MIS 11. The argument that it was of longer duration than other ice free periods is one which has been raised many times before and this manuscript adds little to that. Rather, I would want to motivate the authors to use the ink to put their interesting results in a broader context.

Response: We acknowledge this point and have modified the text accordingly. We have changed the focus from MIS 11 - a decision that was furthermore encouraged by the inclusion of 23 data new points from Beel et al., 2016, which expanded the sampled areas and added data from more landscape types and elevations. Following this inclusion, it was natural to change the focus away from MIS 11 and instead discuss the landscape evolution. However, we have decided to keep a discussion of the most likely exposure-burial history, as we believe it is an important result that these data are fully consistent with exposure-burial histories defined by the $d^{18}O$ record (and hence changes in global ice volume).

I much like the manuscript of Strunk and colleagues and would endorse publication given a broader background description (context), more in-depth discussion of their results against previous literature (it is particularly troublesome that Sugden is missing from the list of references), abandon their MIS 11 focus, and an inclusion of sensitivity tests. The manuscript is well written, is based on a methodology which has been presented in detail elsewhere (Knudsen et al., 2015) and which appears robust, and has some spectacular quantitative results.

Response: We are naturally very pleased with this recommendation and we now provide a broader background description to accommodate a broader scientific audience than in the first submitted manuscript, which was transferred from *Nature Geoscience*. We have abandoned the focus on MIS11, although we keep a very brief discussion of the results in relation to climatic changes in the Arctic. We also include a sensitivity analysis focusing on the presence of till cover (as requested above).

I conclude with a few short comments:

1. Mention of ODP646, Qulleq-1, and ice cores in Fig 1 caption.

Response: Thank you for pointing this out, we have now clarified in the figure caption of Fig. 1 that these points represent pre-LGM, Quaternary data.

2. Line 337, remove "to" between "model" and "the samples".

Response: Thank you, this is now corrected.

3. Figure 2: why is the dashed horizontal red line in-between two red dots? Why does the upper one have inheritance and the lower one not?

Response: The red dashed was placed just at the same altitude as the lower of the two mentioned data points, which was unnecessarily confusing. It should have been placed clearly under both points. In the present revision, we have changed Fig. 2 after the addition of 14 new data points, and we now show one common threshold line for all four sites. (The new data points are based on ^{10}Be and ^{26}Al data published by Beel et al., while this manuscript was under revision).

4. Supplementary line 37: a value "below" (i.e. lower than) the cut-off value means more interglacial not "glacial"!

Response: Thank you very much for pointing out this erroneous wording.

5. Supplementary line 60 and 69-71: UBE 13 is evidently an oddball. Was it not removed from mean values and thus Uummannaq mean value is perhaps based on 5 samples rather than 6?

Response: Yes, the Uummannaq mean value was based on 5 and not 6 samples, as the outlier UBE13 was excluded from the exposure history. We now base the exposure history from Uummannaq on the same 5 samples plus 14 from Beel et al., 2016.

6. Reference quoted and not used in either this ms or in Knudsen et al. (2015): Stroeven, A. P., et al. (2002). "Quantifying the erosional impact of the Fennoscandian ice sheet in the Torneträsk-Narvik corridor, northern Sweden, based on cosmogenic radionuclide data." *Geografiska Annaler* 84A(3-4): 275-287.

Response: We have now included this reference.

Reviewer #2 (Remarks to the Author):

Review of:

One million years of glaciation and denudation history in Greenland
by Strunk et al.

Nature Communications, June 2016

In this manuscript, the authors use a numerical model to quantify rates of denudation and the timing of exposure/burial on the west Greenland landscape. They make use of several existing cosmogenic nuclide datasets (those with ^{10}Be and ^{26}Al analysis on bedrock surfaces) as their data source. They employ an advanced Monte Carlo modeling approach that uses certain inputs (interglacial and glacial erosion rates, oxygen isotope threshold that results in glaciation of the study areas) to infer the most likely scenarios for the study surfaces. The authors conclude that fjord areas have more rapid denudation rates than the highlands, and that there were likely several periods of significant interglacial exposure, in particular MIS11.

I will mention that I am by no means a modeling expert, so the Editor may wish to solicit feedback from someone with more quantitative background than myself who can evaluate the methods thoroughly. Here, I primarily focus on the conceptual framework of the study and the interpretation rather than the details of the model.

Overall, I would like to commend the authors on a job well done. These are interesting questions to pursue, and the authors make good use of pre-existing datasets but take those data to new places. The paper is generally well-written and clear, and the significance of the work is effectively communicated in the introduction.

However, I think the paper could benefit from revision, primarily targeted at increasing the level of depth in the discussion, more effectively communicating the complexities associated with the data and resulting interpretations, and making the writing more accessible to a diverse group of readers. At present, I am not entirely convinced that the paper will be accessible to and/or of interest to a broad audience (although the climatic implications are interesting, and potentially far-reaching). Bringing additional ideas into the discussion, particularly those focused on long-term landscape evolution, may help the paper be of interest to a wider range of readers.

Below, I have detailed several major suggestions as well as a number of minor suggestions. Again, I commend the authors on an interesting study and encourage them to continue expanding its depth and accessibility.

Lee Corbett

Ashley.Corbett@uvm.edu

We would like to thank Lee Corbett for a very thorough and encouraging review, and we have done our utmost best to meet the suggestions for revision.

Big-picture comments:

Knudsen (2015) paper. I was surprised not to see more explicit discussion of the Knudsen (2015) paper here. Although I admittedly lack modeling expertise, the two papers seem like they share some of the same ideas and utilize some of the same data. My impression is that the 2015 paper outlines the technique, while the in-review paper presents findings based on the technique. Perhaps there was additional discussion in the cover letter that I am missing? I don't necessarily see a problem here, I just feel like I am lacking clarity on how the two are different, how the second builds on the first, and how much of the data in the second is new.

Response: The differences between the present manuscript and the paper by Knudsen et al., 2015 are detailed above in response to Reviewer #1. In brief, the mentioned paper by Knudsen et al., 2015 presents the Markov-Chain Monte Carlo modeling approach to paired multi-nuclide data. They do this by applying the model on two data points (from Corbett et al., 2013), thereby showing the ability of their model to integrate exposure and denudation history. The present paper is profoundly different from Knudsen et al., 2015 as we apply the MCMC model approach to all available paired multi-nuclide bedrock data from Greenland. We quantify denudation rates and exposure histories based on 49 data points, which represent the landscape from the present margins of the Greenland Ice Sheet to the coast and enables us to illuminate the landscape evolution across this region.

Background information. I understand you have very limited space here, but is it fair to assume that all readers know what cosmogenic nuclides are? Possibly in Quaternary Science Reviews, but probably not in a Nature Communications piece. Your paper would be more easily accessible to a broad audience if you could add a sentence or two about what these nuclides are and how they are produced. Similarly, a sentence about the difference between warm-based and cold-based ice would be helpful to many readers, as would a sentence about the difference between an erosion rate and a denudation rate. Finally, several terms used in the Methods section could be more effectively explained (see detail below).

Response: Thank you for these suggestions. As the first submitted manuscript was transferred from *Nature Geoscience* with a limit of 1,800 words, we now use the opportunity of less limited space to include more background information to accommodate a broad audience in the current revision (e.g. lines 52-75 and 130-135).

Model assumptions. You have some good discussion of this on Lines 67-79, although personally I would like to see more detail here since communicating the assumptions is so critical for allowing the reader to understand the model (and its limitations). For example, your model does not give you the ability to include periods of interglacial burial of the bedrock surfaces (i.e. by snowfields or till), which is a process often discussed in the Arctic literature. Nor does your model allow you to partially shield a surface, erode a surface episodically, etc. I don't think this is in any way a problem- all models require assumptions- but a more thorough discussion of this could be beneficial.

Response: We agree that understanding the model assumptions is imperative for any reader to be able to assess the model. We have strived to provide more detail about the model assumptions (e.g. lines 272-290). Furthermore, we include a till sensitivity analysis in order to quantify the direct

affect of a till cover on our results. A detailed background for this new addition is also provided in the response to Reviewer #1.

Erosion rates. Nowhere in the paper do you mention the interglacial and glacial erosion rates you used and how you determined them. You describe this in the supplement, but to me it seems like an important piece of information that should be made explicit for the reader. I suggest including a sentence to this effect in the paragraph where you discuss the model; at the least, provide the range of values you used, a brief statement of how you determined those values, and refer the reader to the supplement for more information.

Response: The glacial and interglacial erosion rates are free parameters in the model, i.e. for each sample, we calculate the best fitting erosion rates for these two distinct regimes (glacial and interglacial), as described in line 90-96 and detailed in line 268. An example of the erosion rate results is shown in Fig. S1a-b. As described in Supplementary Information, lines 88-90, we integrate the erosion rates for the two regimes over 1 Myr, to better constrain the total denudation which is shown in Fig. S4.

Samples with "no" inheritance. In a number of places in the text (Lines 89-91, Lines 105-109, and Figure 2), you identify samples that have "no" isotopic inheritance, often in relation to an inferred topographic boundary. To me, this seems overly simplistic. Can you say with certainty that the low-elevation samples are inheritance-free? And how can you confidently distinguish between which samples contain inherited nuclides and which do not? Realistically, all of the samples in your dataset may have at least small amounts of inherited nuclides, but the uncertainties in the single-isotope simple exposure ages and/or the uncertainties on the $^{26}\text{Al}/^{10}\text{Be}$ ratios do not allow you to distinguish between a sample with no inheritance and one with minimal inheritance. I think this is an important point to finesse throughout the paper, especially since it is a limitation of your model and of the technique as a whole.

Response: We agree that all the samples in the dataset realistically contain very small amounts of inheritance, and as you point out, we are not able to definitely distinguish whether samples have inheritance or not, if there is not a boulder date just next to the bedrock sample location. We have now changed the wording to pointing out that all samples, according to our model results, above a certain elevation boundary do contain inheritance, as we can show that they most likely have been ice covered before the LGM.

Denudation rates. One of your main aims is quantifying the difference in denudation rates between fjords and inter-fjord plateaus, and it seems like you have successfully been able to do this. I would be interested to see you take this a step further though, and tie it back to the landscape in west Greenland today. What is the current elevation difference between the fjords and the highlands? When extrapolated over the lifespan of the GrIS, is that at all consistent with the difference in denudation rates you calculated? If not, could there be structural underpinnings to the origin of the fjords? You probably cannot address this question with a high degree of certainty, but I think it would be interesting discussion to include and would make your study less theoretical and more relevant to understanding the modern-day landscape. At present, your claim of understanding the age and origin of the present topography (Lines 51 and 104) seems unrealized.

Response: We agree that elaborating more on the landscape evolution is a very interesting path, and encouraged by both reviewers, we now spend considerably more focus on the processes shaping the landscape, and how our results can help to deepen the understanding of these phenomena (e.g. lines 117-164).

"Chicken and egg" problem. I guess this is an American expression, but hopefully my analogy makes sense. One of your main conclusions is that the fjord areas have higher denudation rates than the intervening highlands, and that this is likely due to the subglacial thermal regime. I agree with your assessment. But are the fjords deep because of the rapid denudation rate? Or is the rapid denudation rate only able to occur because the fjords are deep? Is this a positive feedback loop? In essence, what came first, the topography or the denudation rate? This is a subtlety, but I think it is an interesting question and warrants further exploration. Similar to my point above, this sort of critical discussion will help take your paper from the model to the actual landscape.

Response: We are likewise intrigued by these questions. We attempt to cast more light on the landscape evolution and the controlling factors.

Minor comments:

Line 14 and throughout: I usually see Arctic capitalized.

Response: Thank you, this is now corrected.

Line 19: I would advise against using "all" here; I know there is at least one more bedrock paired-isotope study that I reviewed, and it might be published in advance of your paper.

Response: We have included the dataset by Beel et al. (2016), and we therefore continue to use this phrasing.

Line 24: Do you mean "durations" instead of "periods"? I would use "duration" here to clarify that this is a cumulative amount of time rather than a single period of time.

Response: Thank you for pointing this out. We have rephrased the specific line in the abstract, but we use the suggested nomenclature in later instances.

Lines 26-29: I think you are referring to MIS11 in this sentence, but the usage of "this period" and "this time" is vague.

Response: As we changed focus from MIS 11, we abandoned this sentence, but we have sought to avoid likewise vague references throughout the revised manuscript.

Lines 39-41: As written, the sentence seems to argue that the age of the deposits (lateglacial/Holocene) implies rapid retreat. I think the sentence might need some restructuring. I assume you mean that there are some specific retreat rate estimates?

Response: Thank you for the suggestion, we have made this sentence more clear.

Lines 52-60: The end of this paragraph does not read as smoothly as the rest of your front material. I wonder whether calling out the Svalbard study specifically is worth the space in such a short paper format. You may be better served by condensing this down to a concise sentence focused on the limitations of previous approaches.

Response: We agree with this suggestion and now provide a more general assessment of the current state of long-term exposure/burial dating.

Lines 76-79: I like that you constrain the timing of the most recent deglaciation with independent radiocarbon and/or cosmogenic chronology. For the sake of methodological transparency, it might be helpful to include a table in the supplement of exactly what ages you used and the studies from which they came. This could help future groups, perhaps those with a different model, to use the same assumptions as you did and compare results.

Response: We appreciate your acknowledgement of this approach. However, we chose to use more conservative deglaciation boundaries in the revised manuscript. We mainly did so because the constraints based on radiocarbon/cosmogenic chronology is mostly tied to data points at low elevations. As we seek to investigate the denudation processes in different altitudes, we preferred uniform deglaciation constraints for the whole of each area rather than extrapolating the narrower, secondarily based deglaciation ages from low-lying areas, to high-elevation data points.

Lines 81-82 and Lines 207-208: How did you decide upon this 20ka threshold for which samples you modeled? In the second occurrence, you say that this age is "significantly higher than the age of Holocene deglaciation". But did you actually apply some sort of statistical test, or is this a semi-arbitrary cutoff? If the latter, omit the word "significant".

Response: Thank you for this comment, we agree that the word "significant" should not be used loosely. With a 20kyr cut-off all samples are, within error, older than the age of Holocene deglaciation. For samples closer to the deglaciation age, the model is not able to meaningfully extract an exposure and denudation history without in situ ^{14}C data.

Lines 97-100: The reasoning here seems circular. You say that your calculated denudation rates and calculated exposure/burial durations from previous studies are consistent with one another. But aren't these all based on the same isotopic data, so wouldn't we expect them to be consistent?

Response: We refer to studies based on other isotopic data, which show similar exposure/burial histories, and we are now able to quantify the low denudation rate causing these very old samples.

Lines 107-108: I think you mean "are therefore" instead of "is therefore".

Response: Yes, thank you.

Line 123: See my earlier comment (Line 24) about "durations" versus "periods".

Response: Thank you.

Lines 123-124: It is not clear what the values in parentheses are. Are these first and third quartiles as mentioned in the methods text? I suggest defining this here so that it is easily accessible for the reader.

Response: Yes, the values were the first and third quartiles as mentioned in the methods text. However, as we shift the focus slightly to elaborate more on the landscape denudation and less on the exposure history, we have abandoned pointing out these results.

Line 127: Using the "GDF" acronym here seems unnecessary since you only use the term in this one paragraph. It might be friendlier to the reader to omit it.

Response: We have tried to rephrase the referencing to the glacial debris flows, and now only use the acronym in Fig. 4.

Line 137: Define "Weichselian"; most of the audience probably will not be familiar with this term (or perhaps just use the MIS notation instead).

Response: We now write "Weichselian/Wisconsin glacial period" to clarify this.

Lines 161-162: Use "1 Myr" or "one million years".

Response: Yes, thank you.

Line 209: I think you mean "for" instead of "four".

Response: Yes, thank you.

Line 209 and below (and throughout the supplement as well): What is a "walker"? I am not sure this is a term most of the audience will be familiar with.

Response: We have tried to clarify this.

Line 212: What is the "acceptance ratio"? Similar to the above, this probably will not be familiar to non-modelers.

Response: We have also tried to clarify this.

In-text references: It looks like there are numerous instances where the references are not superscripted. See lines 118, 128, 139, 148, 152, and 155.

Response: We strive to follow the formatting guidelines, and as these references are following a number, we have mentioned them in brackets and not superscripted.

Figure 1. I think this figure could be improved to convey more information. The base imagery here really is not great (and the source of the imagery should be included in the figure caption). Is there another source of imagery that is higher resolution? Additionally, you could contemplate adding some quantitative information (perhaps simple ^{10}Be exposure ages, or maybe dot size as scaled to exposure age?).

Response: We appreciate your suggestions for this figure, and we like the idea of representing the simple ^{10}Be exposure ages from the samples we base our results on. We have attained better imagery and made the figure larger, to make room for more information and the newly added data set from Beel et al., 2016.

Figure 3. It seems odd to me to graph the exposure history on a scale of 0-100% when there are only ever two possible values: 0 or 100. Is there a more intuitive way of symbolizing this for the reader that makes the fact that this is a bimodal system (either complete exposure or complete burial) more clear? Showing this another way might also help the legibility of the plot, since these lines are hard to read because of the underlying colors and the fact that they are close together. Are the plots even necessary, or can you just say in the caption that the study sites are exposed in the green intervals and buried during the blue intervals?

Response: Thank you for your suggestions for this figure. We have kept the 0-100% scale to clarify that each exposure history has its own x-axis. We also kept the plots for each area, so it is possible for the reader to see the differences and similarities in detail. We are convinced that the addition of read lines of high-exposure samples do not worsen the readability of the overall figure, as it is our opinion that we with these plots propose an alternative and more complex exposure history to the conventional interpretation of samples with very high ^{10}Be ages.

All figures. The way in which you denote sub-elements of the figure (a, b, c) is not standard throughout. In some, you lead with the letter; in others, the letter follows the description. Standardizing your notation will make these easier to read.

Response: This is now corrected, so all captions appear with uniform formatting.

Reviewers' comments:

Reviewer #1 (Remarks to the Author):

Review of revised manuscript by Strunk and others for consideration in Nature Communications.

I thank the editor for the possibility to scrutinize the revised manuscript by Astrid Strunk and colleagues, and indeed the authors for an improved manuscript. I am happy with the rebuttal to my previous comments and to see this manuscript printed in Nature Communications.

I offer the following suggestions for minor improvements that reflect some minor mistakes and inconsistencies that remain:

Main manuscript.

Line 52: Why not “at” the pressure melting point rather than “close to”. This would be more consistent with formulations in lines 121-122 (*reach the pressure melting point*).

Line 55: The coupling of the second part of the sentence to the first part with “which explain” is awkward and deserves a rephrasing.

Line 109: I’m unfamiliar with the term “broad trend”, but without statistics I would settle for the much more intuitively “weak trend”.

Line 167: “vary” should be “varies”.

Lines 174-177: I wonder if the authors would like to insert a notion that another explanation for the short cumulative exposure samples would be resetting during some earlier glacial phase and accumulation in-line with other samples during the remaining glacial history? I know that variable erosion rates through time is discussed elsewhere (lines 150-154) as scenarios that cannot be modelled using the MCMC technique, and would be good to refer to reaffirm this point here...

Line 286: remove either “(Ref. 38)” – which is what I would do – or “³⁸”.

Lines 289-291. Sentence “We apply this method... above 20 kyr” is unnecessary repeat of line 279 and can be removed.

Lines 310-313: Although technically the sentence may be grammatically correct, I would suggest that you inserted “is” in-between “samples” and “indistinguishable”.

Line 315: “Fig. 5S” should be “Fig. S5”.

There are many mistakes in the list of references. Here I list the ones I spotted:

Line 325: Add DOI to complete the reference.

Line 330: 525-8 should be 525-528.

Line 331: Remove capitalization of words.

Line 332: Remove “(80-).”.

Line 339: Remove capitalization of words.

Line 342: Remove “(80-).”.

Line 342: 402-5 should be 402-405.

Line 360: Add journal and volume. *Geomorphology* **27**.

Line 376: Change “d” to “ δ ”.

Line 395: Capitalize “Fennoscandian”.

Line 399: 100-3 should be 100-103.

Line 406: Should there be more information after *Computers in Physics* **7**?

Line 407: Remove capitalization of words.

Line 408: Write Science instead of “Sci.”

Line 410: Remove “(80-).”.

Line 412: Capitalize “Greenland”.

Lines 414-415: Remove capitalization of words.

Line 417: Write 10Be instead of 10Be

Line 422: Complete page range (23753-23759).

Line 424: Remove capitalization of words.

Figures and captions:

Lines 435-437: Descriptions of panels 1b and 1c differ unnecessarily as landscape in 1b is also “fjord dissected”.

Line 441: “above” should be “below”.

Figure 2 panels a and b y-axes: altitude should be listed as “m a.s.l.” and not “m. a.s.l.”. Panel a has two samples that are off the x-axis scale. The panel b “crosses” are too hard to read. Consider using another graphic such as a “diamond”.

Figure 3 panel b y-axis: altitude should be listed as “m a.s.l.” and not “m. a.s.l.”.

Supplementary material:

Lines 14-15: Titles don't match with those in the text. Change to “Exposure history” and Denudation rate” – or change in the text.

Figure S1 panels a and b x-axes: Change “Myr-1” to “Myr⁻¹”. Panel d x-axis change $\delta^{18}\text{O}$ to $\delta^{18}\text{O}$ and $\delta^{18}\text{O}$ to $\delta^{18}\text{O}$.

Line 24: 10-12 kyr should be 10-16 kyr.

Figure S2 Red dashed lines (legend and caption, line 30) are not dashed. Y-axis change $\delta^{18}\text{O}$ to $\delta^{18}\text{O}$ and $\delta^{18}\text{O}$ to $\delta^{18}\text{O}$.

Figure S3 Red dashed lines (legend) are not dashed. Y-axes of all three panels: change $\delta^{18}\text{O}$ to $\delta^{18}\text{O}$ and $\delta^{18}\text{O}$ to $\delta^{18}\text{O}$.

Figure S4: Median (black color) cannot be distinguished against the blue background. Use another color (green, yellow?).

Figure S5 panels d-f x-axes: Change " $d^{18}\text{O}$ " to " $\delta^{18}\text{O}$ ".

Line 60: Quote of wrong sample number, wrong study region, and wrong reference. Should be 13-GROR-70 and Uummanaq¹, respectively.

Lines 66-67: Quote of wrong sample number, wrong study region, and wrong reference. Should be 13-GROR-70 and Uummanaq¹, respectively.

Line 69: Should Fig. S3 be Fig. 2?

Lines 69-70: Should Fig. 3 be Fig. S3?

Line 147: should you mention that this is a PhD Thesis?

Supplementary information, excel table.

Change " $d^{18}\text{O}$ Threshold" to " $\delta^{18}\text{O}$ Threshold".

Reviewer #2 (Remarks to the Author):

Second review of:

One million years of glaciation and denudation history in west Greenland

by Strunk et al.

Nature Communications, October 2016

In this manuscript, the authors use a numerical model to quantify rates of denudation and the timing of exposure/burial on the west Greenland landscape. This is my second review of the manuscript, after suggesting significant modification of the first submitted version.

After assessing the new draft and in particular the way in which authors addressed reviewer comments, I think this version marks an improvement over the previous. There is more background and the model parameters have been more thoroughly described. However, there are still some subtleties that could benefit from additional clarification for the reader. The $^{26}\text{Al}/^{10}\text{Be}$ system includes many inherent assumptions, and the authors' model introduces additional assumptions; this is by no means a problem, but these limitations need to be explicit for the reader before he/she can critically assess the study and its results.

Below, I include revisited comments (those that I brought up in my initial review) as well as some new comments. I commend the authors on an interesting and timely study and am happy to see the existing body of Greenland two-isotope data compiled and assessed in a thoughtful manner.

Re-visited comments:

(Referred to by the same heading as in my original review)

Knudsen (2015) paper. I appreciate the authors' clarification on this point. The text is now written more clearly and specifies that the 2015 paper outlines the method whereas the current paper applies the method. The addition of the new data from Beel et al. add to the new data presented in the current paper.

Background information. The authors more effectively provide the reader with the necessary background information in this version of the manuscript. The brief descriptions of cold-based versus warm-based ice, denudation rates, and a few model-related terms are concise and helpful. The longer description of cosmogenic nuclides is detailed and will provide the framework that will allow a diverse audience to understand the approach.

Model assumptions. The addition of discussion regarding model assumptions has improved this version. The authors now provide more detail about model assumptions in the Methods section and several caveats/cautions toward the end of the main text. The addition of the sensitivity analysis is helpful, and I like how you restructured your treatment of the timing of the last deglaciation. However, I think it would be useful to state more of these assumptions up front, since most readers will likely not delve into the Methods section. It seems like several sentences could be

added to the paragraph currently spanning Lines 82-96. For example, you assume that the erosion rate is steady throughout, and do not allow for episodic erosion (e.g. plucking, which is of course a common process subglacially). You do not allow for till cover or snowfields. All of these ideas are included in this paragraph, but only in modeling terms ("full exposure") rather than in real-world terms ("no cover by till or snowfields"). Helping the reader understand the model, its assumptions, its limitations, and its relationship to real-world processes is critical, so I advocate for being particularly explicit in your language here.

Samples with "no" inheritance. The treatment of my concern here is more conservative in this version of the manuscript; the authors now define a threshold above which all samples are assumed to have inheritance, rather than defining a threshold below which samples have "no" inheritance. This seems to be a more robust and realistic approach. However, I would still like to see a few additional sentences regarding this subtlety, especially for the benefit of the uninitiated reader. You of course need to make assumptions for the purposes your model, but the reality is that there is no magic elevation threshold that defines whether inheritance is present or not. I worry that this oversimplification will be misleading; I suggest clarifying that small amounts of inherited nuclides are essentially impossible to identify and likely exist below the threshold as well as above. Additionally, the term "no inheritance" is still used on Lines 128 and 130; reword this to match the other changes you have made along this line (substituting "non-detectable" might work).

Denudation rates and the "chicken and egg" problem. The authors now include a more wide-reaching discussion about long-term landscape development and explore how different denudation rates may lead to the evolution of topography over time. These are intriguing, thought-provoking questions to explore and I think they will make the paper of interest to a wider readership base. I like the addition of the material in Lines 141-154.

Figure 1. This figure is much improved, although still needs fine-tuning. The higher-quality imagery and the simple ^{10}Be exposure ages convey more information than the previous draft. But you should specify in the caption that these are "simple" (or "apparent"- whichever term you prefer) ^{10}Be exposure ages. Also, it looks like the caption for letters b/c are reversed- b is Uummannaq and c is Upernavik. Finally, you have a typo on Line 441 in the caption: "...a $^{26}\text{Al}/^{10}\text{Be}$ ratio above 7.5". Do you mean below?

Figure 3 (now Figure 4). I respect the authors' decision to leave this figure as-is. However, my personal opinion is that it is not very intuitive. I think I finally understand it, but it took me about five minutes of head-scratching, and I worry that it will not be very accessible to a diverse audience. Some simple changes might help; perhaps annotation showing "warmer conditions" and "cooler conditions" (or however you want to word it) as a spectrum from right to left, and I would still advocate for using "Exposure" and "Burial" rather than "100%" and "0%".

New major comments:

"Simple" ^{10}Be exposure ages. This is an important point to make for the reader, since you want him/her to be under no delusion that the ages you present are actual exposure ages. You use the term for the first time on Line 98 but never define it. Specify that a "simple" ^{10}Be exposure age is based on the assumption of no inherited nuclides, no post-glacial erosion, etc, and that the samples you are working with violate these assumptions. Then, make sure you use the "simple" modifier throughout (see my comments about figure captions and axes, which neglect to specify). It seems to me like this would be an easy area for the reader to get misled.

Order of sections. In the manuscript, you discuss denudation rates first and the exposure/burial history second. Have you considered reversing the two? Perhaps I am misunderstanding the structure of your model, but my perception is that you calculate the exposure/burial history with

the 180 threshold first, and then use that history to quantify the denudation rate. If so, presenting them in that order might make more intuitive sense as you build up the reader's understanding of the model.

New minor comments:

Line 33: I suggest softening "virtually blank". I agree that not a lot is known about ice sheet variation, but there is some information from the marine record and a few other sources that you do not want to discount.

Lines 43-45: This sentence is quite specific, doesn't seem like an adequate topic sentence for the paragraph, and doesn't connect logically to the paragraph before. I suggest adding a big-picture topic sentence, something linking repeated ice cover with the development of topography.

Lines 61-63: I would use "nuclear spallation" here, or an equivalent description, rather than "interact", which is vague.

Lines 69-72: Specify for the reader that the paired-isotope technique works because the two have different half-lives and therefore behave differently during burial.

Line 74: Consider "alternating" instead of "interchanging".

Lines 114-116: Specify here that these denudation rates are integrated over both glacial and interglacial periods... i.e. that they encompass both subaerial and subglacial erosion.

Line 135: I would add "increasing" in front of elevation, or specify in some other way so that the reader understands the direction of the relationship.

Lines 135-138: This sentence feels awkwardly-worded.

Lines 160-162: Do you instead mean something like "This cumulative, simulated exposure/burial history is conceptually more advanced than the minimum-limiting exposure and burial durations defined by the simplest pathway to explain a point on the two-isotope diagram..."? The sentence as written seemed incorrect since it was comparing the "sum of exposure" with the "minimum limiting burial duration". I know what you mean, but the wording was confusing.

Lines 167-177: I think you mean "exposure proportion" (i.e. the percentage of exposure out of the total history) rather than "cumulative exposure", right? The former would make more sense, and seems aligned with the fact that you've used percentages.

Lines 173-174: Is there a word missing here? I'm confused about how samples with both long and short exposure result in similar uncertainties. And similar to the above, are you talking about actual durations or proportions?

Line 181: I don't think you've defined what the production ratio is yet. Maybe you could introduce this when you provide background about cosmogenic nuclides?

Lines 181-187: I agree with you here- you can't distinguish between samples that were covered for short durations and those that have been continuously exposed. But tell the reader why this is, since this is a critical piece in understanding your model and its limitations. The $^{26}\text{Al}/^{10}\text{Be}$ system can't detect short burial durations- why not? And how short is "short"?

Line 195: Sample GU110 is from Upernavik, not Sisimiut.

Line 283: The word "removal" is odd here. Consider using "decay/erosion" instead, or other words that accurately describe the processes at play.

Line 286: I would like to see the sea-level high-latitude production rates (and the $^{26}\text{Al}/^{10}\text{Be}$ production ratio) you used, rather than just the reference to the source.

Figure 2. See my comment for Figure 1- specify here and throughout that these are simple/apparent ^{10}Be exposure ages.

Figure 3. In the caption, you say that the lower black line marks "steady erosion under constant exposure". This isn't worded correctly. That lower line is actually the end-points of an infinite number of different erosion rate scenarios, right? So the envelope above marks the territory where any scenario involving constant exposure and any erosion rate will land. The lower black line is itself not a single scenario. I know this is a subtlety, but you don't want the reader thinking that a constantly-exposed but eroded surface will follow that trajectory.

Reviewers' comments:

Reviewer #1 (Remarks to the Author):

Review of revised manuscript by Strunk and others for consideration in Nature Communications.

I thank the editor for the possibility to scrutinize the revised manuscript by Astrid Strunk and colleagues, and indeed the authors for an improved manuscript. I am happy with the rebuttal to my previous comments and to see this manuscript printed in Nature Communications.

I offer the following suggestions for minor improvements that reflect some minor mistakes and inconsistencies that remain:

We would like to thank Arjen Stroeven for highly constructive, helpful, and thorough comments. We appreciate the suggestions for improvements and have incorporated all of the suggested changes.

Main manuscript.

Line 52: Why not “at” the pressure melting point rather than “close to”. This would be more consistent with formulations in lines 121-122 (reach the pressure melting point).

Line 55: The coupling of the second part of the sentence to the first part with “which explain” is awkward and deserves a rephrasing.

Line 109: I’m unfamiliar with the term “broad trend”, but without statistics I would settle for the much more intuitively “weak trend”.

Line 167: “vary” should be “varies”.

Lines 174-177: I wonder if the authors would like to insert a notion that another explanation for the short cumulative exposure samples would be resetting during some earlier glacial phase and accumulation in-line with other samples during the remaining glacial history? I know that variable erosion rates through time is discussed elsewhere (lines 150-154) as scenarios that cannot be modelled using the MCMC technique, and would be good to refer to reaffirm this point here...

Line 286: remove either “(Ref. 38)” – which is what I would do – or “38”.

Lines 289-291. Sentence “We apply this method... above 20 kyr” is unnecessary repeat of line 279 and can be removed.

Lines 310-313: Although technically the sentence may be grammatically correct, I would suggest that you inserted “is” in-between “samples” and “indistinguishable”.

Line 315: “Fig. 5S” should be “Fig. S5”.

There are many mistakes in the list of references. Here I list the ones I spotted:

Line 325: Add DOI to complete the reference.

Line 330: 525-8 should be 525-528.

Line 331: Remove capitalization of words.

Line 332: Remove “(80-).”.

Line 339: Remove capitalization of words.

Line 342: Remove “(80-).”.

Line 342: 402-5 should be 402-405.

Line 360: Add journal and volume. Geomorphology 27.

Line 376: Change “d” to “ δ ”.

Line 395: Capitalize “Fennoscandian”.

Line 399: 100-3 should be 100-103.

Line 406: Should there be more information after Computers in Physics 7?

Line 407: Remove capitalization of words.

Line 408: Write Science instead of “Sci.”

Line 410: Remove “(80-).”.

Line 412: Capitalize “Greenland”.

Lines 414-415: Remove capitalization of words.

Line 417: Write ^{10}Be instead of 10Be

Line 422: Complete page range (23753-23759).

Line 424: Remove capitalization of words.

Figures and captions:

Lines 435-437: Descriptions of panels 1b and 1c differ unnecessarily as landscape in 1b is also “fjord dissected”.

Line 441: “above” should be “below”.

Figure 2 panels a and b y-axes: altitude should be listed as “m a.s.l.” and not “m. a.s.l.”.

Panel a has two samples that are off the x-axis scale. The panel b “crosses” are too hard to read. Consider using another graphic such as a “diamond”.

Figure 3 panel b y-axis: altitude should be listed as “m a.s.l.” and not “m. a.s.l.”.

Supplementary material:

Lines 14-15: Titles don’t match with those in the text. Change to “Exposure history” and Denudation rate” – or change in the text.

Figure S1 panels a and b x-axes: Change “Myr-1” to “Myr⁻¹”. Panel d x-axis change $\delta^{18}\text{O}$

to $\delta^{18}\text{O}$ and $\delta^{18}\text{O}$ to $\delta^{18}\text{O}$.

Line 24: 10-12 kyr should be 10-16 kyr.

Figure S2 Red dashed lines (legend and caption, line 30) are not dashed. Y-axis change $\delta^{18}\text{O}$ to $\delta^{18}\text{O}$ and $\delta^{18}\text{O}$ to $\delta^{18}\text{O}$.

Figure S3 Red dashed lines (legend) are not dashed. Y-axes of all three panels: change $\delta^{18}\text{O}$ to $\delta^{18}\text{O}$ and $\delta^{18}\text{O}$ to $\delta^{18}\text{O}$.

Figure S4: Median (black color) cannot be distinguished against the blue background. Use another color (green, yellow?).

Figure S5 panels d-f x-axes: Change "d18O" to " $\delta^{18}\text{O}$ ".

Line 60: Quote of wrong sample number, wrong study region, and wrong reference. Should be 13-GROR-70 and Ummannaq1, respectively.

Lines 66-67: Quote of wrong sample number, wrong study region, and wrong reference. Should be 13- GROR-70 and Ummannaq1, respectively.

Line 69: Should Fig. S3 be Fig. 2?

Lines 69-70: Should Fig. 3 be Fig. S3?

Line 147: should you mention that this is a PhD Thesis? Supplementary information, excel table. Change "d18O Threshold" to " $\delta^{18}\text{O}$ Threshold".

Best wishes, Arjen Stroeven

Reviewer #2 (Remarks to the Author):

Second review of:

One million years of glaciation and denudation history in west Greenland
by Strunk et al.

Nature Communications, October 2016

In this manuscript, the authors use a numerical model to quantify rates of denudation and the timing of exposure/burial on the west Greenland landscape. This is my second review of the manuscript, after suggesting significant modification of the first submitted version.

After assessing the new draft and in particular the way in which authors addressed reviewer comments, I think this version marks an improvement over the previous. There is more background and the model parameters have been more thoroughly described. However, there are still some subtleties that could benefit from additional clarification for the reader. The $^{26}\text{Al}/^{10}\text{Be}$ system includes many inherent assumptions, and the authors' model introduces additional assumptions; this is by no means a problem, but these limitations need to be explicit for the reader before he/she can critically assess the study and its results.

Below, I include revisited comments (those that I brought up in my initial review) as well as some new comments. I commend the authors on an interesting and timely study and am happy to see the existing body of Greenland two-isotope data compiled and assessed in a thoughtful manner.

We would like to thank Lee Corbett for a highly constructive and helpful review. We have strived to accommodate all the suggestions, and below we provide a point-by-point response to the suggestions for improvements.

Re-visited comments:
(Referred to by the same heading as in my original review)

Knudsen (2015) paper. I appreciate the authors' clarification on this point. The text is now written more clearly and specifies that the 2015 paper outlines the method whereas the current paper applies the method. The addition of the new data from Beel et al. add to the new data presented in the current paper.

Background information. The authors more effectively provide the reader with the necessary background information in this version of the manuscript. The brief descriptions of cold-based versus warm-based ice, denudation rates, and a few model-related terms are concise and helpful. The longer description of cosmogenic nuclides is detailed and will provide the framework that will allow a diverse audience to understand the approach.

We are pleased that the revised manuscript is considered an improvement and that the text is now more accessible to a diverse audience. In the present revision, we have done our best to give more background information on the method in general and to clarify the assumptions of the model.

Model assumptions. The addition of discussion regarding model assumptions has improved this version. The authors now provide more detail about model assumptions in the Methods section and several caveats/cautions toward the end of the main text. The addition of the sensitivity analysis is helpful, and I like how you restructured your treatment of the timing of the last deglaciation. However, I think it would be useful to state more of

these assumptions up front, since most readers will likely not delve into the Methods section. It seems like several sentences could be added to the paragraph currently spanning Lines 82-96. For example, you assume that the erosion rate is steady throughout, and do not allow for episodic erosion (e.g. plucking, which is of course a common process subglacially). You do not allow for till cover or snowfields. All of these ideas are included in this paragraph, but only in modeling terms (“full exposure”) rather than in real-world terms (“no cover by till or snowfields”). Helping the reader understand the model, its assumptions, its limitations, and its relationship to real-world processes is critical, so I advocate for being particularly explicit in your language here.

We have added a few sentences in lines 100-107, in which we more explicitly mention the model assumptions and limitations with a more “real-world” terminology.

Samples with “no” inheritance. The treatment of my concern here is more conservative in this version of the manuscript; the authors now define a threshold above which all samples are assumed to have inheritance, rather than defining a threshold below which samples have “no” inheritance. This seems to be a more robust and realistic approach. However, I would still like to see a few additional sentences regarding this subtlety, especially for the benefit of the uninitiated reader. You of course need to make assumptions for the purposes your model, but the reality is that there is no magic elevation threshold that defines whether inheritance is present or not. I worry that this oversimplification will be misleading; I suggest clarifying that small amounts of inherited nuclides are essentially impossible to identify and likely exist below the threshold as well as above. Additionally, the term “no inheritance” is still used on Lines 128 and 130; reword this to match the other changes you have made along this line (substituting “non-detectable” might work).

We agree that there is no magic elevation threshold defining an exact boundary for inheritance. To avoid this misconception, we have been careful to use the term “non-detectable inheritance” throughout the manuscript. In lines 136-142 we explicitly address that this threshold is a generalization.

Denudation rates and the “chicken and egg” problem. The authors now include a more wide-reaching discussion about long-term landscape development and explore how different denudation rates may lead to the evolution of topography over time. These are intriguing, thought-provoking questions to explore and I think they will make the paper of interest to a wider readership base. I like the addition of the material in Lines 141-154.

We are very glad to hear that the revised manuscript might interest a wider audience, and we enjoyed the opportunity to expand the discussion of long-term landscape development.

Figure 1. This figure is much improved, although still needs fine-tuning. The higher-quality imagery and the simple ^{10}Be exposure ages convey more information than the previous draft. But you should specify in the caption that these are “simple” (or “apparent”- whichever term you prefer) ^{10}Be exposure ages. Also, it looks like the caption for letters b/c are reversed- b is Uummannaq and c is Upernavik. Finally, you have a typo on Line 441 in the caption: “...a $^{26}\text{Al}/^{10}\text{Be}$ ratio above 7.5”. Do you mean below?

The suggested improvements and mentioned mistakes in the caption are corrected, and the text on the figure should be more readable in this revised revision.

Figure 3 (now Figure 4). I respect the authors' decision to leave this figure as-is. However, my personal opinion is that it is not very intuitive. I think I finally understand it, but it took me about five minutes of head-scratching, and I worry that it will not be very accessible to a diverse audience. Some simple changes might help; perhaps annotation showing "warmer conditions" and "cooler conditions" (or however you want to word it) as a spectrum from right to left, and I would still advocate for using "Exposure" and "Burial" rather than "100%" and "0%".

We are grateful to the reviewer for these ideas. We have added an arrow to explain the $d^{18}O$ value (warm/cold conditions), a legend to explain that the blue/green colors are burial/exposure periods, and changed the x-axes to be Exp. and Bur. instead of 100% and 0%. We also changed the (c) title to be more reader-friendly. We hope that the changes will make the figure more accessible.

New major comments:

"Simple" ^{10}Be exposure ages. This is an important point to make for the reader, since you want him/her to be under no delusion that the ages you present are actual exposure ages. You use the term for the first time on Line 98 but never define it. Specify that a "simple" ^{10}Be exposure age is based on the assumption of no inherited nuclides, no post-glacial erosion, etc, and that the samples you are working with violate these assumptions. Then, make sure you use the "simple" modifier throughout (see my comments about figure captions and axes, which neglect to specify). It seems to me like this would be an easy area for the reader to get misled.

Thank you for pointing this out, we now explain "simple exposure age" (lines 120-123) and are careful to use this term when applicable throughout the paper.

Order of sections. In the manuscript, you discuss denudation rates first and the exposure/burial history second. Have you considered reversing the two? Perhaps I am misunderstanding the structure of your model, but my perception is that you calculate the exposure/burial history with the ^{18}O threshold first, and then use that history to quantify the denudation rate. If so, presenting them in that order might make more intuitive sense as you build up the reader's understanding of the model.

We appreciate the suggestion. However, we did not change the structure of the manuscript, as the model does not calculate either of the parameter values before the other. The model picks out a set of values for all four parameters at once for each iteration, and changes all four parameters at once for the next iteration. We find the existing structure of the paper to be more logical with respect to the discussion of landscape

evolution in west Greenland.

New minor comments:

Line 33: I suggest softening “virtually blank”. I agree that not a lot is known about ice sheet variation, but there is some information from the marine record and a few other sources that you do not want to discount.

Thank you for pointing this out - we have changed the wording accordingly (line 34).

Lines 43-45: This sentence is quite specific, doesn't seem like an adequate topic sentence for the paragraph, and doesn't connect logically to the paragraph before. I suggest adding a big-picture topic sentence, something linking repeated ice cover with the development of topography.

Thank you for this suggestion, we now open this paragraph with a more “big-picture” sentence (44-46).

Lines 61-63: I would use “nuclear spallation” here, or an equivalent description, rather than “interact”, which is vague.

We have followed the suggestion and use a more precise expression (65).

Lines 69-72: Specify for the reader that the paired-isotope technique works because the two have different half-lives and therefore behave differently during burial.

We now explicitly specify the basis of the two-isotope system (76-77).

Line 74: Consider “alternating” instead of “interchanging”.

The text is changed accordingly (81).

Lines 114-116: Specify here that these denudation rates are integrated over both glacial and interglacial periods... i.e. that they encompass both subaerial and subglacial erosion.

The text is changed as suggested, and we now explain more clearly that the denudation rate is an integration of subglacial and subaerial erosion (96-107).

Line 135: I would add “increasing” in front of elevation, or specify in some other way so that the reader understands the direction of the relationship.

The text is changed accordingly (164).

Lines 135-138: This sentence feels awkwardly-worded.

We have omitted the last part of the sentence, which did not follow the logic of the

preceding text (lines: 164-166). Thanks for pointing this out.

Lines 160-162: Do you instead mean something like “This cumulative, simulated exposure/burial history is conceptually more advanced than the minimum-limiting exposure and burial durations defined by the simplest pathway to explain a point on the two-isotope diagram...”? The sentence as written seemed incorrect since it was comparing the “sum of exposure” with the “minimum limiting burial duration”. I know what you mean, but the wording was confusing.

Thank you for this suggestion, we have changed the text accordingly (188-191).

Lines 167-177: I think you mean “exposure proportion” (i.e. the percentage of exposure out of the total history) rather than “cumulative exposure”, right? The former would make more sense, and seems aligned with the fact that you’ve used percentages.

We have changed the text accordingly (194).

Lines 173-174: Is there a word missing here? I’m confused about how samples with both long and short exposure result in similar uncertainties. And similar to the above, are you talking about actual durations or proportions?

We have tried to make it more clear that the low uncertainties are associated with i) samples with low cumulative exposure and ii) samples with high cumulative exposure, i.e. the samples in the “extreme” ends of the range of Al/Be ratios. We have also clarified that we mean the proportions (202-204).

Line 181: I don’t think you’ve defined what the production ratio is yet. Maybe you could introduce this when you provide background about cosmogenic nuclides?

We now introduce how the system is based on different production rates and half-lives in lines 76-79.

Lines 181-187: I agree with you here- you can’t distinguish between samples that were covered for short durations and those that have been continuously exposed. But tell the reader why this is, since this is a critical piece in understanding your model and its limitations. The $^{26}\text{Al}/^{10}\text{Be}$ system can’t detect short burial durations- why not? And how short is “short”?

We have clarified this point as suggested (216-223).

Line 195: Sample GU110 is from Upernavik, not Sisimiut.

We have changed the text accordingly (230).

Line 283: The word “removal” is odd here. Consider using “decay/erosion” instead, or other words that accurately describe the processes at play.

We have changed the text accordingly (303).

Line 286: I would like to see the sea-level high-latitude production rates (and the $^{26}\text{Al}/^{10}\text{Be}$ production ratio) you used, rather than just the reference to the source.

We have added the sea-level high-latitude production rates as suggested. The $^{26}\text{Al}/^{10}\text{Be}$ production ratio is evident from these two values (306).

Figure 2. See my comment for Figure 1- specify here and throughout that these are simple/apparent ^{10}Be exposure ages.

We have changed the figure and caption accordingly.

Figure 3. In the caption, you say that the lower black line marks “steady erosion under constant exposure”. This isn’t worded correctly. That lower line is actually the end-points of an infinite number of different erosion rate scenarios, right? So the envelope above marks the territory where any scenario involving constant exposure and any erosion rate will land. The lower black line is itself not a single scenario. I know this is a subtlety, but you don’t want the reader thinking that a constantly-exposed but eroded surface will follow that trajectory.

Yes – that is absolutely correct. The former text was misleading - we have changed the figure and caption accordingly.

REVIEWERS' COMMENTS:

Reviewer #2 (Remarks to the Author):

Third review of:

One million years of glaciation and denudation history in west Greenland
by Strunk et al.

Nature Communications, November 2016

In this manuscript, the authors use a numerical model to quantify rates of denudation and the timing of exposure/burial on the west Greenland landscape. This is my third review of the manuscript.

After assessing the latest draft, I am satisfied with how the authors have incorporated reviewer feedback. I found the model assumptions and limitations to be more clearly articulated, which will make the manuscript more accessible for a more diverse audience. I also appreciated the greater level of care and caution with which the cosmogenic data were discussed, including more explicit treatment of the issues of inheritance, "simple" exposure ages, and age assumptions that were raised in previous reviews.

I commend the authors on an interesting study, which in my opinion is now ready for publication.